
# The heavy precipitation event of 14–15 October 2018 in the Aude catchment: A meteorological study based on operational numerical weather prediction systems and standard and personal observations

Olivier Caumont[1], Marc Mandement[1], François Bouttier[1], Judith Eeckman[1,2],
Cindy Lebeaupin Brossier[1], Alexane Lovat[1,3], Olivier Nuissier[1], and Olivier Laurantin[3]

[1]CNRM, Université de Toulouse, Météo-France, CNRS, Toulouse, France
[2]IMFT, Université de Toulouse, CNRS, Toulouse, France
[3]Observing Systems Division, Météo-France, Toulouse, France

**Correspondence:** Olivier Caumont (olivier.caumont@meteo.fr)

**Abstract.** The case of the heavy precipitation event on 14 and 15 October 2018 which has led to severe flash flooding in the Aude watershed in south-western France is studied from a meteorological point of view using deterministic and probabilistic numerical weather prediction systems, as well as a unique combination of observations from both standard and personal weather stations. This case is typical of Mediterranean heavy precipitation events due to its classic synoptic situation and its

quasi-stationary convective precipitation that regenerates continuously, but with some peculiarities such as the presence of a former hurricane and a pre-existing cold air mass close to the ground.

It is shown that the positive Mediterranean sea surface temperature anomaly may have played an aggravating role in the amount of precipitation that poured into the Aude basin. On the other hand, soil moisture does not seem to have played a significant role. A study of rainfall forecasts shows that the event had limited predictability, in particular given the small size

of the watersheds involved. It is shown that the stationarity of precipitation, whose estimation benefits from data from personal stations, is linked to the presence near the ground of a trough and a strong potential virtual temperature gradient, the stationarity of both of which is highlighted by a combination of observations from standard and personal stations. The forecast that comes closest to the rainfall observations contains the warmest, wettest and fastest low-level jet and also simulates near the ground a trough and a marked boundary between cold air in the west and warm air in the east, both of which are stationary.

## 1 Introduction

Heavy precipitation events (HPEs) are common in coastal regions bordering the Mediterranean (e.g., Ricard et al., 2012). These events regularly cause flash floods with tragic consequences, as the catchment areas affected are small and therefore react very quickly (Sivapalan et al., 2002; Berne et al., 2004; Creutin et al., 2009). The mechanisms at the origin of heavy rainfall involve the Mediterranean Sea, the relief and a favourable synoptic situation. The combination of these three ingredients can, however,

lead to a multitude of different configurations. The interactions between relief and atmospheric circulation can be particularly complex. In addition to the simple uplift of a moist, unstable low-level jet by the relief, other more complex mechanisms can





take place, such as the creation of convergence zones through deflection or channelling of this jet by the relief. Pre-existing thunderstorms can also have an effect on the initiation of new convective cells, particularly through the creation of a density current that can initiate deep convection at its leading edge. The most extreme events may be due to the stationary nature of the

precipitation or its long duration.

In order to anticipate these events and reduce their impact, it is necessary to set up forecasting chains ranging from numerical rainfall forecasting to impacts on property and people, ideally by propagating uncertainties along this chain. For some time now, initiatives have been advancing in the integration of forecast chains, giving rise to the prospect of improvements in the effectiveness of these warning systems. For instance, the European COST 731 Action 'Propagation of uncertainty in

advanced meteo-hydrological forecast systems' promoted the use of integrated flood forecasting models and systems (Rossa et al., 2011). Later, the DRIHM and DRIHM2US projects (Parodi et al., 2017) aimed to facilitate research on hydrometeorological forecast chains by developing a distributed infrastructure integrating numerical weather prediction systems, hydrological models and hydraulic models. National initiatives such as PICS (Payrastre et al., 2019a) aim to enhance and transfer these advances from research to operations by designing and evaluating integrated forecast chains capable of anticipating the impacts

of flash floods within a few hours, notably through interactions between various scientific teams (meteorologists, hydrologists, hydraulic engineers, economists, sociologists) and operational stakeholders (civil security, local authorities, insurance companies, hydropower companies, transport network operators). At the international level, the High-Impact Weather (HIWeather; Jones et al., 2014) project, under the World Weather Research Programme (WWRP) of the World Meteorological Organization (WMO), promotes international collaborative research to significantly increase resilience to extreme weather around

the world by improving forecasts for periods ranging from a few minutes to two weeks and enhancing their communication and usefulness in social, economic and environmental applications. Concerning extreme hydrometeorological events, uncertainties in numerical rainfall prediction remain particularly important and constitute a bottleneck for improving the efficiency of flood forecast chains (e.g., Hally et al., 2015). A better understanding of heavy rainfall events is necessary to identify the weaknesses of numerical weather prediction systems and thus focus efforts on the most critical aspects of these systems, be

they observations, model parameterizations, resolution or others. This was, for example, the objective of the Hydrological in the Mediterranean Experiment First Special Observation Period (Ducrocq et al., 2014) that took place in the north-western Mediterranean in autumn 2012.

Here we are interested in a case of a Mediterranean HPE that entailed dramatic consequences. In the night of 14 to 15 October 2018 the Aude, and to a lesser extent, the Hérault and the Tarn departments, in the south of France, were affected by

heavy rainfall. A regenerative multicellular convective system affected the region. This led to flash floods that caused the death of fifteen people and injured 75 people; 7000 homes were flooded and 24000 people suffered material damages for a cost of several hundred million euros. During the event, the villages of Pezens and Conques-sur-Orbiel were completely evacuated. Victims and damage were particularly concentrated in the area on the left bank of the Aude, between the Orbiel and Fresquel Rivers, but not necessarily on these rivers (Villegailhenc is located on the ungauged Trapel stream). This case was chosen

because it has had particularly dramatic consequences, but also because it is atypical, since the 2018 floods took place nineteen





years after one of the major precipitating episodes recorded in the same region, the episode of 12–13 November 1999 (Nuissier et al., 2008; Ducrocq et al., 2008). In 2018, the maximum rainfall occurred about 30 km west of the one observed in 1999.

The objective of this article is to examine the behaviour of operational models and identify the ingredients that led to the 14-15 October 2018 disaster. Among the innovative tools used to carry out this undertaking, storm-scale ensemble forecasting
enables correlations between processes to be identified and data from personal weather stations make it possible to observe, at unprecedented spatial resolutions, the near-ground meteorological signatures of the precipitating system and its environment responsible for the flash flood.

Section 2 describes the model and observational meteorological data used in this study. The case of 14-15 October 2018 is then described in section 3 using global analyses and radar observations. The performance of the different forecasting systems
is then studied in section 4, in particular in order to relate the presence or absence of ingredients according to whether the forecasts simulate the event correctly or not. Section 5 builds on the previous one to investigate the links between rainfall and other meteorological signatures. Conclusions are presented in section 6.

## 2  Meteorological data

In this section, the numerical weather prediction (NWP) systems used in this study are first presented. Radar observations and
weather station observations are then described.

### 2.1  NWP systems

The suite of Météo-France NWP systems is used in this study. This includes ARPEGE, a global spectral model with variable resolution and a four-dimensional variational (4D-Var) data assimilation system (Courtier et al., 1991). Besides ARPEGE, different NWP systems based on AROME are used, such as AROME-France (Seity et al., 2011; Brousseau et al., 2016), its
nowcasting version AROME-NWC (Auger et al., 2015), and its ensemble prediction version AROME-EPS (Bouttier et al., 2012; Raynaud and Bouttier, 2016; Bouttier et al., 2016). AROME features a non-hydrostatic dynamical model core inherited from ALADIN (Members of the ALADIN international team, 1997; Termonia et al., 2018), detailed moist physics shared with the Meso-NH model (Lafore et al., 1998; Lac et al., 2018), and an associated three-dimensional variational (3D-Var) data assimilation scheme (Brousseau et al., 2011). AROME runs at horizontal resolutions at which deep convection is largely
resolved. There is therefore no parameterization of deep convection activated for the AROME-based models used in this study.

AROME-France is used by Météo-France for short-term (up to 2 days) regional forecasts over France at a horizontal resolution of 1.3 km and 90 vertical levels. Analyses are performed every hour and longer forecasts are run every 6 hours.

AROME-NWC is a configuration of AROME especially designed for nowcasting purposes. It runs every hour and provides 6 h forecasts with sub-hourly outputs (15 min). Its forecasts are available within 35 min after the analysis time. AROME-NWC
mainly assimilates data from weather stations and radars, the latter being particularly relevant for nowcasting (e.g., Rossa et al., 2010).



AROME-EPS is a 12-member ensemble based on perturbations of the AROME-France model at a resolution of 2.5 km in 2018. The AROME-EPS system is updated every six hours.

## 2.2 Radar observations

The operational weather radar network in Metropolitan France was composed of 30 radars in October 2018. In this study, the French operational base reflectivity, i.e. measured at the lowest elevation angle of the radar, mosaicked from these 30 radars is used. It has a 1 km × 1 km spatial resolution and a 5 min temporal resolution with reflectivities ranging from –9 dBZ to 70 dBZ with a 0.5 dBZ step. For every pixel in the mosaic, the maximum base reflectivity from radars distant by 180 km or less is taken. If the pixel is distant by more than 180 km to every radar, the maximum base reflectivity of radars at a distance between 95 180 km and 250 km is taken. More details on the French radar network are given by Figueras i Ventura and Tabary (2013).

During the event, the radar located in Opoul suffered several down times, at 20:45, 21:05, 21:30 UTC on 14 October and between 21:55 UTC and 06:05 UTC on 15 October. As a result, reflectivity and derived products have been mainly underestimated around this radar at these times.

## 2.3 Surface observations

Surface observations result from the combination of standard weather stations (SWSs), which are Météo-France operational weather stations sampling atmospheric parameters at a time step of 1 min on the one hand, and crowdsourced personal weather stations (PWSs), on the other hand. The PWS time series of mean sea level pressure (MSLP), temperature and relative humidity are processed following the method presented by Mandement and Caumont (2020). Gridded analyses of surface pressure, mean sea level pressure, temperature, relative humidity, and virtual potential temperature derived from observations near the ground 105 are built at a 5 min time step and 0.01° resolution in latitude and longitude. For MSLP and relative humidity, the gridding method used is the inverse distance weighting (IDW) with a power factor of two. For surface pressure and temperature, the method used is a linear regression over the altitude followed by the IDW with a power factor of 2 of the residuals. Virtual potential temperature fields are built from the previous fields. Details are given by Mandement and Caumont (2020).

For rainfall, ANTILOPE quantitative precipitation estimate (QPE) algorithm (Laurantin, 2008, 2013) is used. It combines 110 radar data (see previous section) and rain gauge data. The estimation of stratiform and convective rainfall in ANTILOPE is based on a spatial-interpolation geostatistical method: kriging with external drift (KED), which allows to take into account an auxiliary spatial variable (here the radar QPE) to interpolate point values (here rain gauge observations). It only involves observations within a radius of 100 km around the point to be calculated. The KED is applied on the one hand to the large-scale rain-gauge accumulations with the large-scale radar QPE as an auxiliary variable and on the other hand to the small-scale 115 rain-gauge accumulations with the small-scale radar QPE as an auxiliary variable. The ANTILOPE QPE is the sum of the two estimates. ANTILOPE QPE has thus a spatial structure close to that of the radar but with values adjusted according to the observed rainfall totals. In particular, the total accumulation of a pixel containing a rain gauge is equal to the total measured by the rain gauge.





For operations, ANTILOPE is run in real time, as well as in delayed time to benefit from more rain gauge data. Total rainfall
is estimated at fifteen-minute or hourly intervals over France at a spatial resolution of 1 km. Here, the delayed-time version of
ANTILOPE is used as a reference. In this study, the inclusion of PWS data in the ANTILOPE algorithm in addition to SWS
data is tested and evaluated (see section 3.2). This inclusion is expected to be all the more beneficial to the ANTILOPE QPE
as the radar suffered failures during the event.

## 3    Case description

An overview of the case is first given from the point of view of the meteorological context. The ARPEGE model is used for
this. The case is then described from a hydrometeorological point of view, this time using standard and personal observations.

### 3.1    Meteorological context

During the week of 10–15 October 2018, several coastal regions neighbouring the northwestern Mediterranean basin were
concerned by intense precipitating episodes. The Balearic Islands were the first impacted by very heavy downpour, in the
night of 9 October 2018, that have ravaged the eastern coast of Mallorca, killing 13 people and causing considerable material
damage (Lorenzo-Lacruz et al., 2019). A few days later, the southern regions of France (especially the Aude department) were
concerned, in turn, by heavy precipitation. The meteorological context during the period was characterised by slow-evolving
patterns including high values of geopotential at 500 hPa centred over Central Europe and surges of geopotential anomalies
over Western Atlantic and Iberian Peninsula. These meteorological conditions are typical synoptic scale patterns favouring
heavy precipitation over the north-western part of the Mediterranean (Nuissier et al., 2008; Ricard et al., 2012; Duffourg et al.,
2016, among others).

Furthermore, the remnants of hurricane Leslie, over the Atlantic ocean, could also have enhanced these severe weather
conditions over western Europe. Indeed, after having made two transitions into a subtropical storm, Leslie was a large, erratic
and long-lived tropical cyclone in the Atlantic which finally became a powerful hurricane-force post-tropical system just west
of the coast of Portugal. The lower (upper) level synoptic situation over Northeastern Atlantic ocean and Western Europe is
presented in figs. 1 and 2, from 18:00 UTC on 12 October until 06:00 UTC on 15 October 2018. Leslie strengthened over
the Northeastern Atlantic and reached a peak intensity with sustained winds of $150 \, \mathrm{km \, h^{-1}}$ and a minimum central pressure
of 969 hPa, on 12 October (not shown). Leslie then accelerated north-eastwards on 13 October while gradually weakening
and interacting with an upper-level trough which forced its transition into an extratropical cyclone (fig. 1a,b). The remnants
of Leslie impacted Western Europe in two ways : (i) a direct impact over land where winds reaching up to $175 \, \mathrm{km \, h^{-1}}$ were
recorded in Portugal and (ii) very moist low-level air masses were advected downstream of the Leslie-trough merged system,
and fed a quasi-stationary cold front over southwestern France in the night of 14 October, generating heavy thunderstorms
and leading to flash flooding in that area. Figure 1d shows that the anomalies of specific humidity calculated between 0 and
3 km altitude exceeded by a factor 3 the standard deviation over a region between the Balearic Islands and Gibraltar straits.
This very high water vapour might be related to warm anomalies at the sea surface already present over the North Atlantic at





mid-latitudes spring and the hurricane season. A significant surface low pressure deepening occurred when the remnants of Leslie and the associated upper level geopotential anomalies crossed through and moved downstream of the Iberian peninsula's steep orography (fig. 2c,d). A strengthening low-level flow then establishes over the Mediterranean Sea and conveys very high low-level moisture contents towards coastal regions of Southeastern France.

The beginning of autumn 2018 was indeed characterized by a positive Sea Surface Temperature (SST) anomaly over the European Atlantic shelf and the Western Mediterranean Sea, as shown in fig. 3 by the OSTIA analyses (Donlon et al., 2012). The SST anomaly was more marked in the South-Western Mediterranean area with values up to 4 °C and persisted until 15 October. It appears also to correspond to the area with large anomalies in specific humidity as highlighted by the ARPEGE analyses (fig. 1d-f). Using the ARPEGE forecasts, the areas of high evaporation can be highlighted (fig. 4). During the after-
noon, large evaporation took place in the gulf of Cadiz and the Alboran Sea, then intensified in the evening and night as moving slowly north-eastwards to the Balearic Islands. These progression and intensification are related to large winds at low levels due to the remnants of Leslie, that favour the heat and vapour extraction from the warm sea. Strong evaporation also took place south-west of Sardinia, and locally along the French coastal area, related to the rapid south-easterly/easterly low-level flow (fig. 1d-f).

### 3.2 Hydrometeorological description of the event

During the night of 13 to 14 October, precipitation first occurs on the south-eastern flank of the Massif Central. As the rainfall intensity increases, the affected area extends to the piedmont plains of the Hérault department and eventually to the Mediter-ranean Sea. In the morning, convective cells develop over the sea and are advected north-westwards towards the Hérault department's coast. The convective cells over the sea disappear after noon. Then orographic precipitation persists over the
south-eastern flank of the Massif Central, whilst slowly decaying (fig. 5a).

During the day, some convective cells cross the Pyrenees in a south-south-westerly flow and produce rain showers over the south of the Pyrenees-Orientales department (fig. 5a). The flow becomes south-easterly in the late afternoon and convective cells appear from 18:30 UTC around the eastern tip of the Pyrenees, which are advected north-westwards towards the Montagne Noire, which forms the southern tip of the Massif Central (fig. 5b). Convective cells keep developing in the same south-easterly
flow for hours (fig. 5c), until they are shifted north-westwards from 05:00 UTC on 15 October. The Hérault department is again affected by rainfall for hours until 13:00 UTC on 16 October 2018 (fig. 5d).

Two maps of accumulated rainfall observed during 48 hours are shown in fig. 6. Figure 6a shows the accumulated rainfall observed by the Météo-France operational product ANTILOPE which blends radar data and conventional rain gauges. This product is hereafter referred to as "SWS ANTILOPE". Figure 6b shows the cumulative rainfall observed when manually quality
checked, supplementary PWS observations are included in the ANTILOPE product. This product is hereafter referred to as "SPWS ANTILOPE". A validation performed on 34 independent automatic and manual rain gauges over six watersheds of the Aude region shows a better agreement of the rainfall accumulation computed by adding PWS data in RMSE, compared to the operational rainfall accumulation (fig. 7). As a consequence, this new QPE is used in the following as a reference.


**Figure 1.** Infrared satellite images superimposed to ARPEGE analysis in terms of mean sea level pressure (dotted lines, hPa), horizontal winds at 925 hPa (barbs, knots) and anomalies of specific humidity (solid lines, g kg$^{-1}$) exceeding 3 and 4 standard deviation, respectively, between 0 and 3 km height and over the domain shown in the Figure.


**Figure 2.** Same as fig. 1 but for geopotential height at 500 hPa (solid lines, m s$^{-1}$), geopotential height (coloured areas, km) and horizontal winds (barbs, m s$^{-1}$) along the 2 PVU surface.

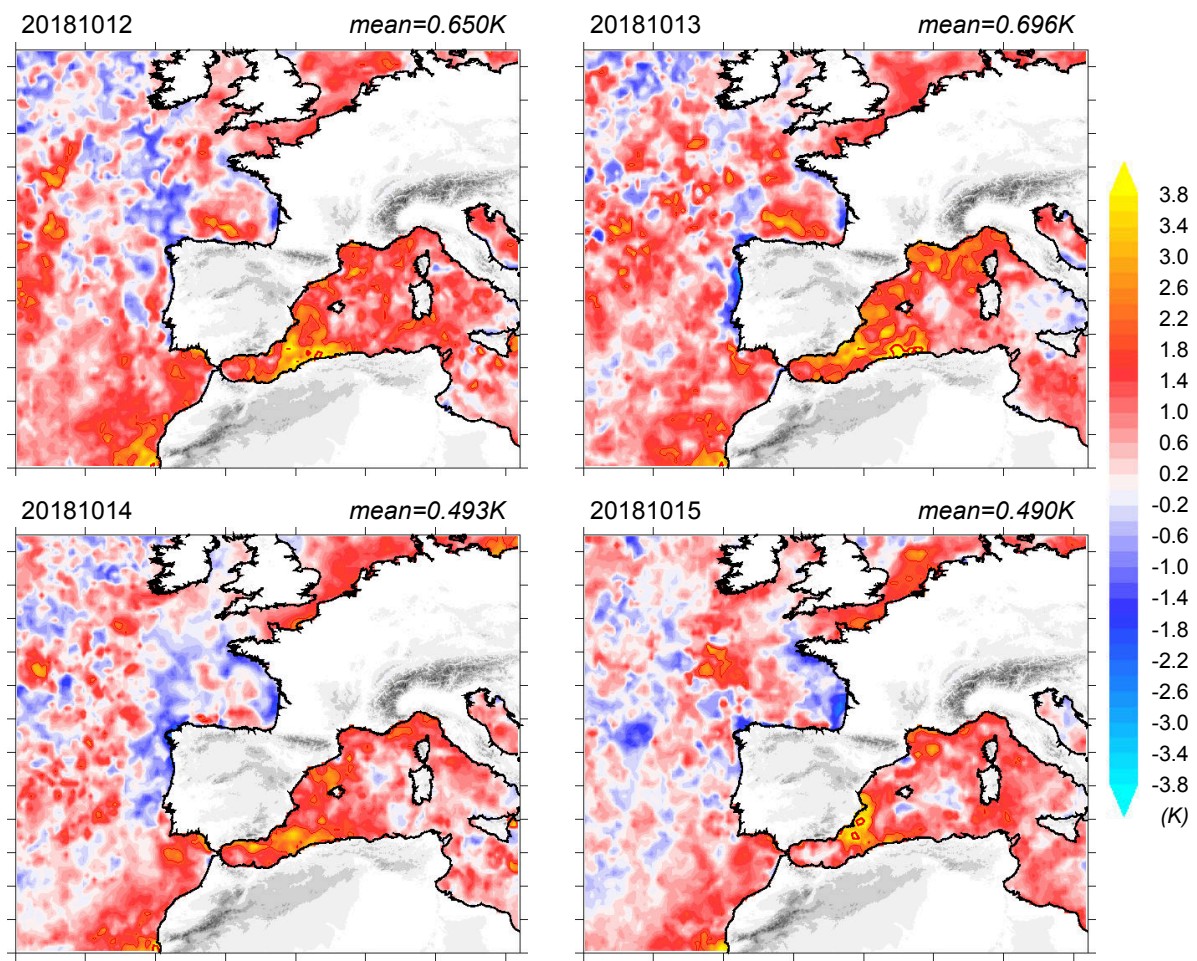

**Figure 3.** Daily SST anomalies (K) for 12, 13, 14 and 15 October 2018, from the OSTIA analyses.

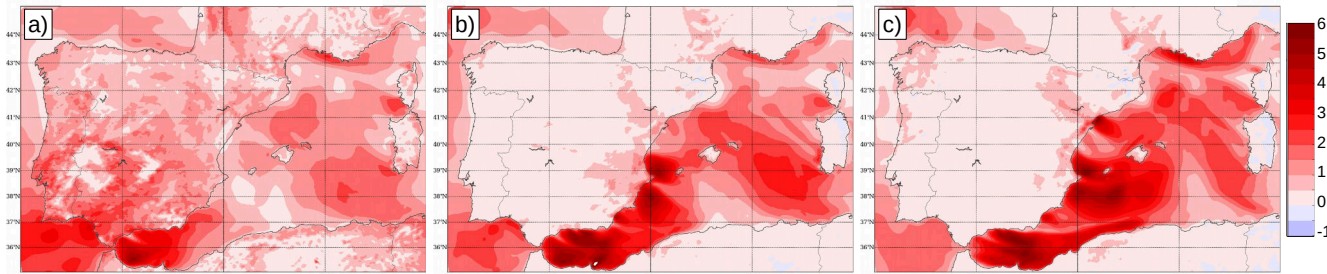

**Figure 4.** 6-hour cumulative surface evaporation ($kg\,m^{-2}$) from successive ARPEGE forecasts: (a) at 18:00 UTC on 14 October (forecast basis: 14 October 12:00 UTC), (b) at 00:00 UTC on 15 October (forecast basis: 14 October 18:00 UTC), and (c) at 06:00 UTC on 15 October (forecast basis: 15 October 00:00 UTC).

**Figure 5.** Composite radar reflectivity (dBZ) (a) at 16:30 UTC on 14 October; (b) at 18:55 UTC on 14 October; (c) at 21:15 UTC on 14 October; (d) at 09:55 UTC on 15 October.






**Figure 6.** 48 h rainfall accumulation from (a) SWS ANTILOPE (b) SPWS ANTILOPE superimposed with rain gauges accumulations in mm between 06:00 UTC on 14 October and 06:00 UTC on 16 October. Major river basins are indicated in bold black. The 48-hour period has been chosen to include manual Météo-France rain gauges to evaluate both products (in the "Validation" group).


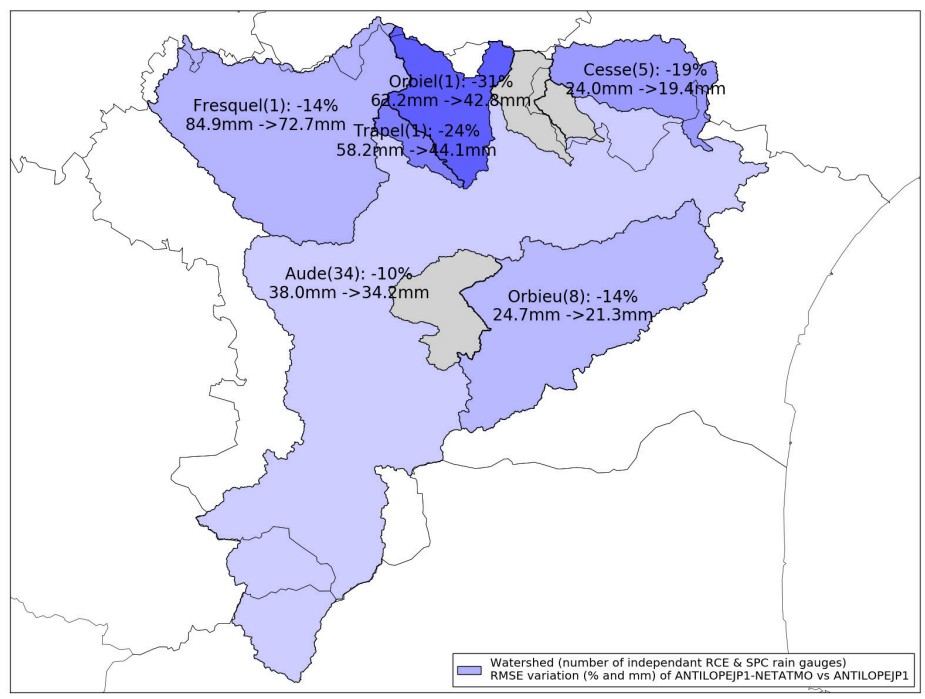

**Figure 7.** RMSE variation (% and mm) between SPWS ANTILOPE and SWS ANTILOPE over validation rain gauges in each watershed of the Aude basin. The name of the watershed and the number of validation rain gauges (in parenthesis) are indicated. Grey watersheds indicate that no validation rain gauges are available.

Rainfall accumulations exceeding 300 mm are observed in the Aude department between 06:00 UTC on 14 October and
06:00 UTC on 16 October (fig. 6b). The rainfall pattern extends north-westwards from the eastern tip of the Pyrenees, reaching a maximum in the piedmont of the Montagne Noire, to the north-western boundary of the Tarn department. Rainfall intensity was very strong in places on 15 October with, for example, 57.6 mm in 1 hour measured at Caunes-Minervois at 04:00 UTC, 55.5 mm in 1 hour measured at Trèbes at 03:00 UTC and Lézignan-Corbières at 06:08 UTC. But the accumulations over periods of between 3 and 12 hours were the most remarkable, with return periods reaching one hundred years, observed for
example at Trèbes with 295.5 mm in 12 hours, Arquettes-en-Val with 212.1 mm in 12 hours and Carcassonne with 113.0 mm in 6 hours.

These large rainfall amounts in a widespread area have caused several rivers to burst their banks from the early morning of 15 October on. In Trèbes and Moussoulens, both on the Aude River, the maximum water peak heights, were close to historical records. In Pezens on the Fresquel River, the water peak height, which occurred at 09:00 UTC, exceeded historical records and
the village had to be evacuated. In Ventenac-en-Minervois and Saint-Marcel, both on the Aude River, the peak water heights, reached at 17:00 UTC and 20:00 UTC, respectively, exceeded historical records.





**Table 1.** Characteristics of the main studied catchments and outlets. Their locations are given in fig. 8 from west to east. Discharge peaks are provided by the French HYDRO data bank when the outlets are monitored or come from post-event surveys conducted in the framework of HyMeX (Payrastre et al., 2019b; Lebouc et al., 2019). See data availability section at the end of this article.

| River | Outlet | Name | Area (km$^2$) | Maximum discharge observed/estimated (m$^3$ s$^{-1}$) (hour & date) |
|---|---|---|---|---|
| Fresquel | Pezens | O1 | 733 | 173 (07:00 15 Oct.) |
| Lauquet | Saint-Hilaire | O2 | 173 | ≈ 800 (HyMeX) |
| Trapel | Villedubert | O3 | 19 | ≈ 125 (HyMeX) |
| Orbiel | Bouilhonnac | O4 | 239 | 481 (03:00 15 Oct.) |
| Argent double | La Redorte | O5 | 108 | 169 (07:00 15 Oct.) |
| Ognon | Pépieux | O6 | 47.1 | 76.9 (07:00 15 Oct.) |
| Orbieu | Villedaigne | O7 | 748 | 1000 (13:00 15 Oct.) |
| Cesse | Mirepeisset | O8 | 257 | 460 (11:00 15 Oct.) |
| Aude | Moussan (near Cuxac-d'Aude) | O9 | 4838 | 1650 (20:00 15 Oct.) |

The maximum rainfall amounts were located in the Trapel catchment, on the west side of the Orbiel River, on the north-east of the Fresquel River and on many rivers in the south of Carcassonne. In the early hours of 15 October, the first increases of flows are observed for the left bank tributaries of the Aude River (table 1). Because of the rainfall intensity and the orientation of the precipitation patterns, the high surface runoff and the basins concentration times, there was only a short period of time
between the start of the rain and the start of the flood peaks. For example, the flow of the Orbiel River rose by 420 m$^3$ s$^{-1}$ in only three hours. The return periods of the Orbiel and Lauquet Rivers were estimated between 100 and 250 years, and those of the Orbieu and Cesse Rivers are around 40 years. The flood of the Aude River in Trèbes is mainly explained by the exceptional and successive water inflows from the Trapel, Orbiel, and Fresquel Rivers and upstream tributaries.

Wet soils are known to prevent precipitation from infiltrating, resulting in higher runoff regardless of other environmental conditions, and can therefore lead to more severe flash floods (Grillakis et al., 2016). The Safran-Isba-Modcou chain (Habets et al., 2008) provides estimations of the superficial soil water content on an 8 km resolution grid, at the daily time step. Figure 9 presents the soil wetness index (SWI) on daily average for the day before the beginning of the rain event (i.e. from 06:00 UTC on 12 October to 06:00 UTC on 13 October), together with the daily reference according to a 1981–2010 time series. A band
of relatively high soil moisture is present over the mid-altitude terrains in the north-east of the region of interest. This band corresponds to the precipitation event that happened on 10 October over the region. However, this wet area is mainly located outside of the Aude catchment and most of the soils within the Aude catchment were neither particularly dry nor saturated. In particular, table 2 presents the spatial average SWI per basin, for the Aude River at Cuxac and three of its sub-basins mainly impacted. Apart from the Orbiel River basin at Bouilhonnac, whose SWI on 12 October is 0.26 % higher than the daily




**Figure 8.** Stream network of the Aude River at Moussan (O9) and boundaries of the studied catchments. The red squares correspond to the outlets. The stars indicate the location of the victims of the event.

**Table 2.** Soil wetness index (SWI) on 12 October, on average per basin, together with the daily normal according to the 1981–2010 records, for the Aude River at Cuxac and three of its mainly impacted sub-basins.

|  | SWI on 12 October (%) | 1981–2010 daily normal (%) |
|---|---|---|
| Aude River at Cuxac | 0.54 | 0.45 |
| Fresquel River at Pezens | 0.57 | 0.53 |
| Orbiel River at Bouilhonnac | 0.63 | 0.37 |
| Orbieu River at Villedaigne | 0.54 | 0.54 |





**Figure 9.** Soil wetness index (SWI) provided by SIM, on a 24-hour average from 06:00 UTC on 12 October to 06:00 UTC on 13 October, and the daily reference according to the 1981–2010 records.





reference, the average values of SWI for the presented catchments do not significantly differ from their daily reference. So soil
moisture was not particularly large and probably did not play a significant role in the flood of 14 and 15 October 2018.

## 4  Performance of operational NWP systems

NWP systems such as those based on AROME can realistically forecast precipitation within one hour to two days. They are an
important component of flash-flood warning systems, but are also tools for understanding the meteorological phenomena that
cause heavy precipitation. In this section, the capabilities of the deterministic and ensemble versions of AROME to forecast
heavy precipitation on 14 and 15 October 2018 are evaluated.

### 4.1  Deterministic regional NWP systems (AROME-France and AROME-NWC)

The objective of deterministic systems is to optimize computational resources, the use of observations, and resolution in order
to produce forecasts that are as close as possible to reality with sufficient lead time. These systems have variable refresh
periods depending on the maximum forecast term. We focus first on AROME-France which produces long forecasts (up to
one to two days) every 3 or 6 hours. Figure 10 shows the average rainfall over three critical watersheds predicted by different
runs of AROME-France. It shows that AROME-France can at the same time overestimate (run of 18:00 UTC on 14 October
for the Lauquet River in flood rise), underestimate (flood rise of the Trapel River) or correctly estimate (run of 18:00 UTC on
14 October for the Orbiel River in flood rise) the average rainfall per catchment area. This is due to spatial discrepancies in
rainfall forecasts compared to observations, which is a well-known problem for convective rainfall over small watersheds (e.g.,
Vincendon et al., 2011, and references therein). Forecast errors are larger than the observational errors that can be roughly
inferred from the two ANTILOPE estimates. Another observation is that the rainfall forecasts are not stable from one run to
the next. The runs from 12:00 UTC and 18:00 UTC on 13 October, and from 12:00 UTC on 14 October give totals justifying
an alert, while the other runs give lower totals, even if the precipitation structure is generally respected (not shown). This
complicates the work of forecasters and crisis managers, and shows that the event is of limited predictability.

Nowcasting systems provide forecasts with lead times of less than a few hours that can be useful to face the unpredictable
nature of very intense and localized rainfall events (Stensrud et al., 2009; Sun et al., 2014; Stensrud et al., 2013; Yussouf
and Knopfmeier, 2019). A few hours of anticipation time could afford great benefits to flood warning and intervention of
emergency services. Figure 11 shows time series of 15 min rainfall forecasts with the AROME-NWC nowcasting system. As
for AROME-France, we notice that AROME-NWC overestimates, underestimates and correctly estimates basin rainfall in the
same basins as AROME-France, respectively, but with less spread around the observation than AROME-France in general,
because the forecasts are shorter.

Like AROME-France, AROME-NWC has difficulty predicting the start of the event. Figure 12 shows for example that
rainfall amounts are strongly underestimated between 21:00 and 22:00 UTC on 14 October. In contrast, fig. 13 shows that
AROME-NWC forecasts rainfall rather well between 01:00 UTC and 02:00 UTC on 15 October, with a spatial shift of only a
few tens of kilometres, but sufficient to affect the wrong catchments.

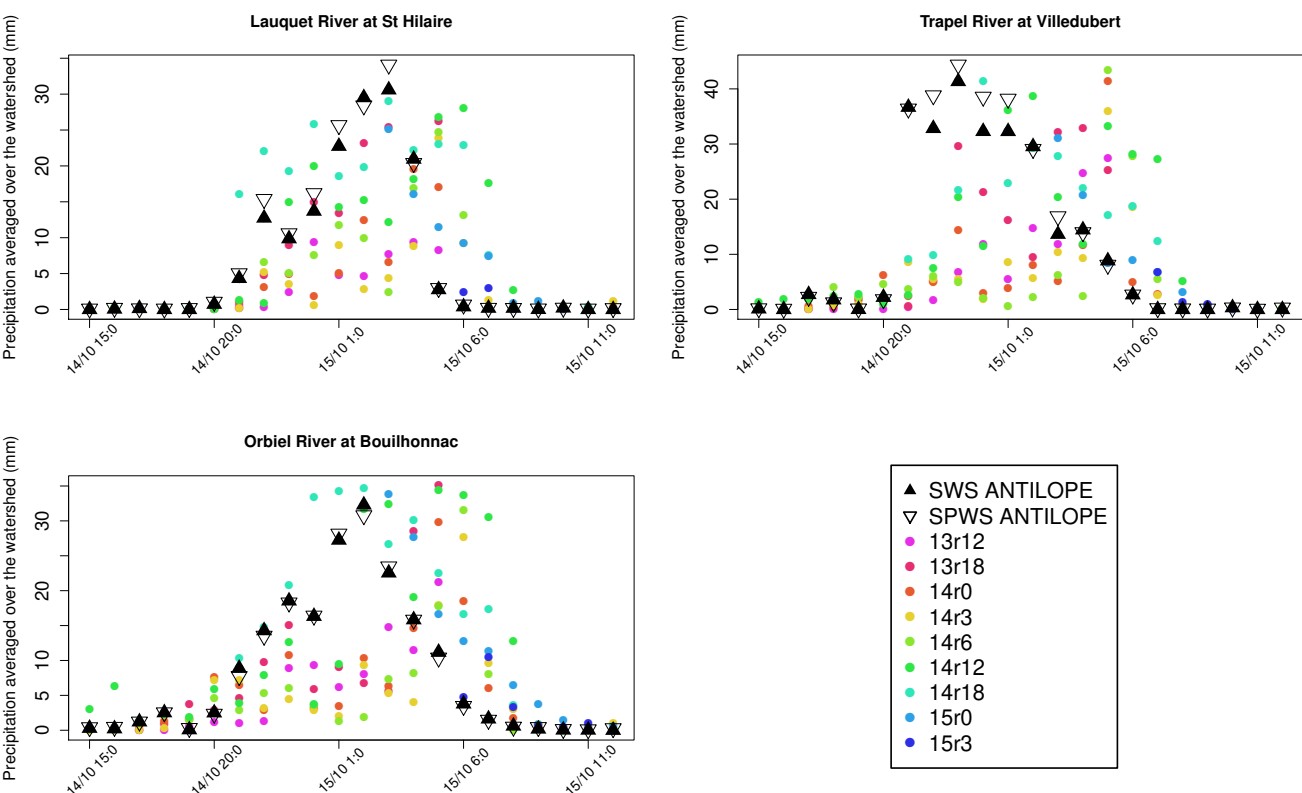

**Figure 10.** Time series of average hourly rainfall per watershed (in mm), observed by ANTILOPE and predicted by various runs of AROME-France. The nomenclature associated with the coloured dots is as follows: XrY corresponds to the run of AROME-France starting at Y UTC on X October 2018.

## 4.2 AROME ensemble prediction system (AROME-EPS)

Ensemble prediction is widely regarded as a key tool for anticipating flood risks, because it enables one to identify warning levels with a predefined false alarm ratio (Cloke and Pappenberger, 2009). When there is a significant risk of damage to life

and property, a common means of expressing the forecast precipitation intensity is by issuing high quantiles of the forecast probability density function. For a well-calibrated ensemble forecast, the 85 % percentile will express a warning level that elicits about 6 times more false alarms than non-detections (i.e. events that occurred without warning). Thus, this percentile is a good approximation of the rain values that a forecaster would issue in order to minimize the consequences of underforecasting such a catastrophic event, without risking issuing so many false alarms that he/she would lose credibility. A more rigorous approach

to weather warning optimization is provided by the so-called economic value score (e.g. Zhu et al., 2002), but this tool is not used here because, being a statistical tool it would require many independent cases to be applied. Here, we simply check how


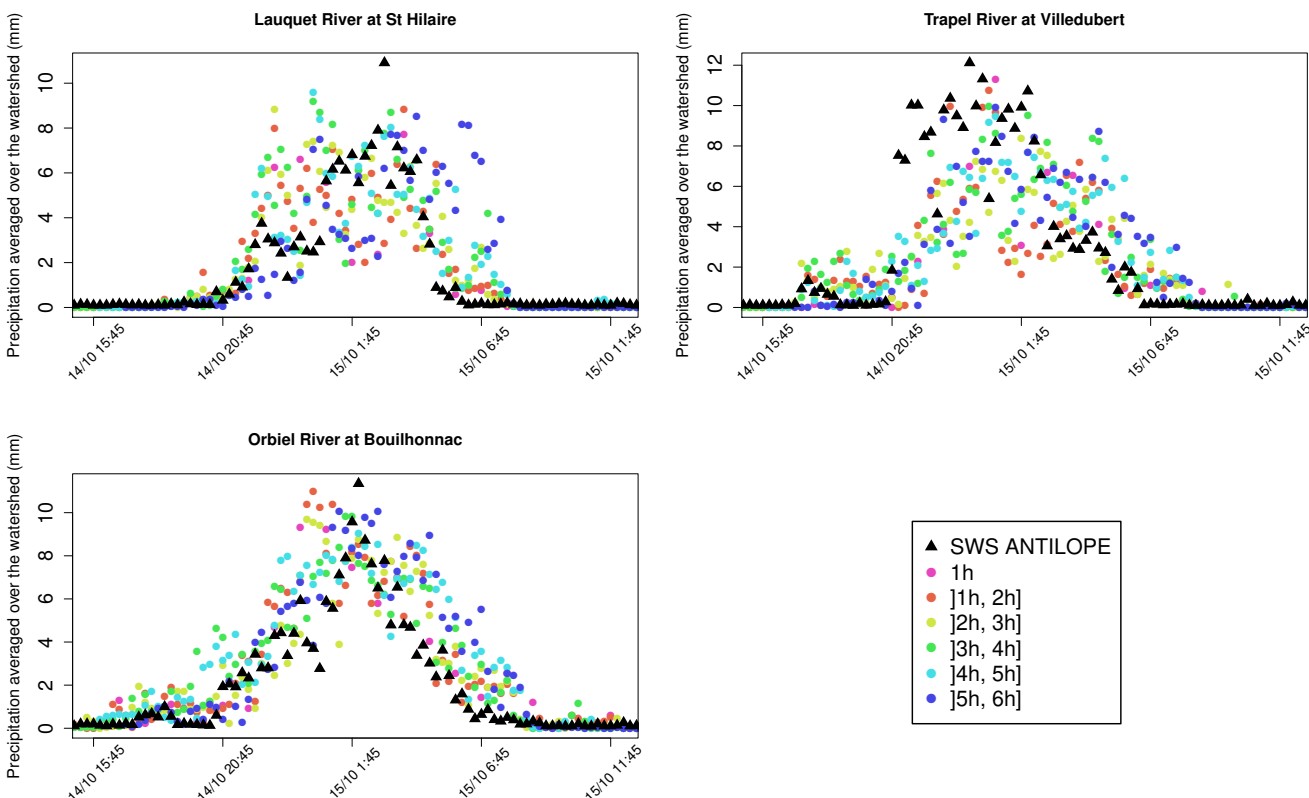

**Figure 11.** Time series of average 15 min average rainfall (in mm) averaged over three watersheds from observation data (black triangles), and from the successive forecasts of AROME-NWC (coloured circles) between 15:00 UTC on 14 October 2018 and 12:00 UTC on 15 October 2018. Colors correspond to forecast lead times.

informative the ensemble prediction could have been to a forecaster that relied on the 85 % percentile to estimate the flooding risk.

Figure 14 shows the three most recent 85 % percentile forecasts for the 12 h precipitation accumulated from 20:00 UTC

on 14 October to 08:00 UTC on 15 October. This period is nicknamed 'period of interest', it encompasses most of the heavy rainfall that led to flooding. One can see that 180 mm accumulations in the area were predicted by the last two ensembles, which were available up to 8 hours before the start of the period, but not before. The last two ensembles predicted rainfall in excess of 180 mm with a location error of the order of 10 km, which is impressively accurate from a meteorological point of view. It remains to be seen, however, whether this would have been precise enough to warn about catastrophic flooding at the

right locations, if these rainfall forecasts had been used to drive hydrological models. Some watersheds in the area are so small that a location error of even a few kilometres could conceivably make the difference between a successful or a failed flood warning.

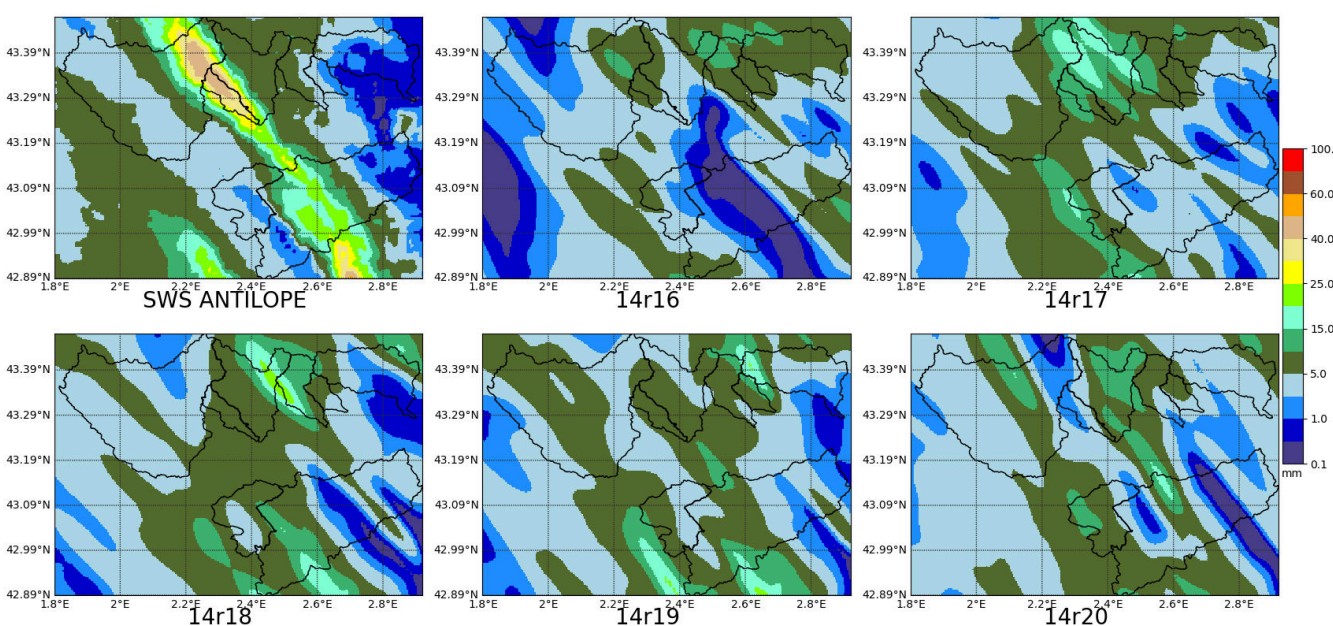

**Figure 12.** Hourly rainfall accumulation (in mm) between 21:00 and 22:00 UTC on 14 October, observed by ANTILOPE and predicted by successive runs of AROME-NWC. The same nomenclature as in fig. 10 is used to refer to AROME-NWC runs.

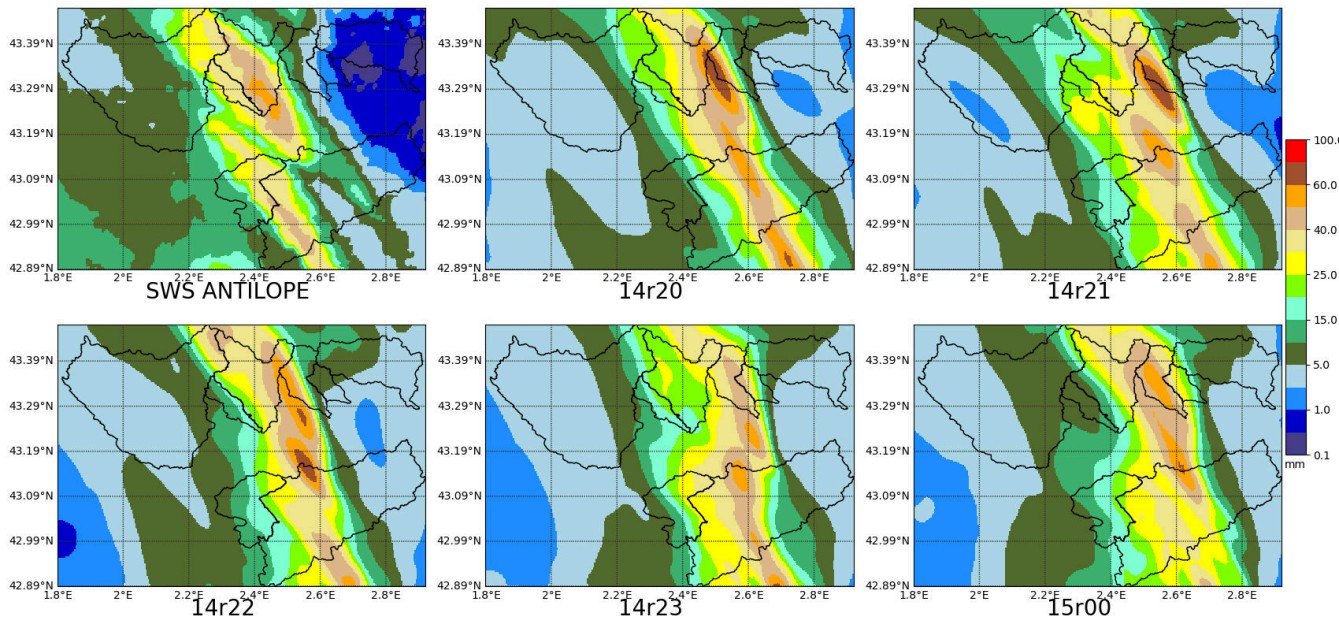

**Figure 13.** Same as in fig. 12, but between 01:00 UTC and 02:00 UTC on 15 October.




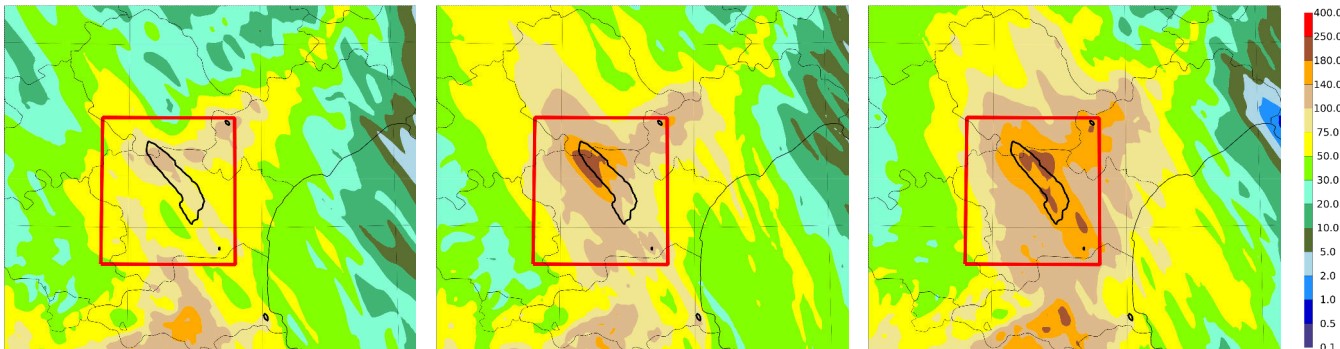

**Figure 14.** 85 % percentile of the predicted 12 h rainfall accumulation over the period of interest from 20:00 UTC on 14 October to 08:00 UTC on 15 October. From left to right, respectively: AROME-EPS forecasts based on 03:00, 09:00 and 15:00 UTC 14 October analyses i.e. 17, 11 and 5 h before the period of interest. These forecasts were available for use approximately 3 hours after the analysis time. The black contour delineates areas where SWS ANTILOPE observed rain exceeds 180 mm. The red rectangle indicates the integration area used to rank the ensemble members.

Figure 15 illustrates the time behaviour of the ensemble forecasts with respect to the observations. The blue areas show the probability distribution for two 12-member ensembles. They show that all members predicted heavy precipitation, with a substantial spread: the ratio between the 85 % and 15 % percentiles (which delineate the bulk of the forecast distribution) is nearly 3 for the 03:00 UTC based forecast, and of the order of two for the 15:00 UTC one, which was the best available in real time. As already shown by fig. 14, the 15:00 UTC forecast is also the most accurate in terms of intensity. The highest quantiles are shifted in time with respect to the observed precipitation peak, by about 4 hours before and after the most intense observed values (with intensities over 8 mm h$^{-1}$). The time evolution of the forecasts is illustrated by the 3 most rainy members only, for the sake of readability. It shows that there is great variability in the precipitation timing: on time scales shorter than 3 hours, there is little apparent relationship between the forecast and observed variations of intensities. As seen in section 3.2, large rainfall accumulations result from a succession of intense convective cells; our time series indicate that, although there is predictive value in the precipitation accumulated over at least 6 hours which what matters most for flood prediction, the individual convective events that cause shorter intense precipitation do not seem predictable beyond a few hours.

In summary, fig. 15 indicates that, although the higher quantiles of the ensembles provided an accurate representation of the overall event in space, time and intensity, there still was substantial uncertainty: with the available ensemble prediction system, precipitation could not have been forecast with less than a 50 % uncertainty on intensity, and about 3 hours in terms of timing.

## 5   Relationship between rainfall and other meteorological mesoscale features

To study the relationship between rainfall and other meteorological mesoscale features, members of the AROME-EPS run starting at 15:00 UTC on 14 October, which was the latest available run before the event, are ranked in order of proximity to rainfall observations. Departures are quantified by computing the Fractions Skill Score (FSS, Roberts (2008)), using the Python





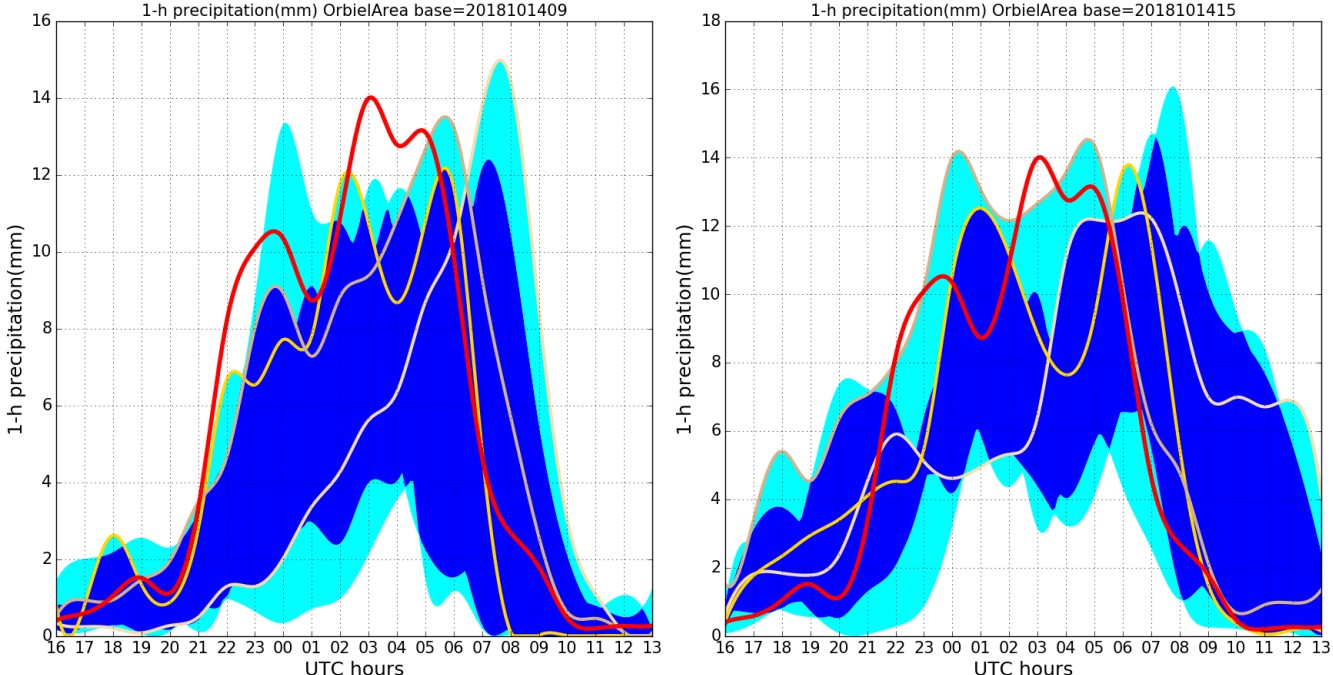

**Figure 15.** Time series of hourly precipitation over the area of interest (depicted in red in fig. 14). Left and right panels, respectively: AROME-EPS forecasts based on the 09:00 and 15:00 UTC analyses on 14 October. Red curve: SWS ANTILOPE observation. Yellow: forecasts from three rainiest members of each ensemble. Light blue area: ensemble minimum and maximum forecasts. Dark blue areas: interval between the 15 % and 85 % percentiles.

code of Faggian et al. (2015). FSS is computed for each member for a range of rainfall thresholds from 0 to the maximum rainfall observed in 12 hours by step of 1 mm. Spatial tolerances between 2 (0.02°) and 64 (0.64°) grid points are considered. The members are ranked in fig. 16 by decreasing mean FSS over all thresholds and all spatial tolerances over the red area shown
in fig. 14: the highest the FSS, the closest the member to the observation. During the period of interest, the three ensemble members closest to the observations are ensemble members with indices 7, 3 and 6. Only member 7 is able to forecast more than 250 mm in 12 hours, and 3 members (1, 10, 11) forecast less than 140 mm in 12 hours over the area that received more than 180 mm (fig. 16).

Figure 17 gives some insight about the physical causes of precipitation ensemble spread. Rainfall (from the 15:00 UTC
run) has been integrated over the period of interest and averaged inside the rectangular domain depicted in red. In terms of this measure, the rainiest ensemble members had indices 7, 6, 3 with respective average 12 h rainfall of 126, 104, and 95 mm. These values are all exceptionally high and associated with local values over 200 mm. These rainiest members are in this case also the three with the highest FSS (but not in the exact same order). Figure 17 shows the average anomaly of these 3 members with respect to the ensemble mean. In other words, it gives clues about the correlation between the forecast rainfall and other
forecast parameters, similarly to Ancell (2016) who used correlation maps from ensembles to investigate the mechanisms of


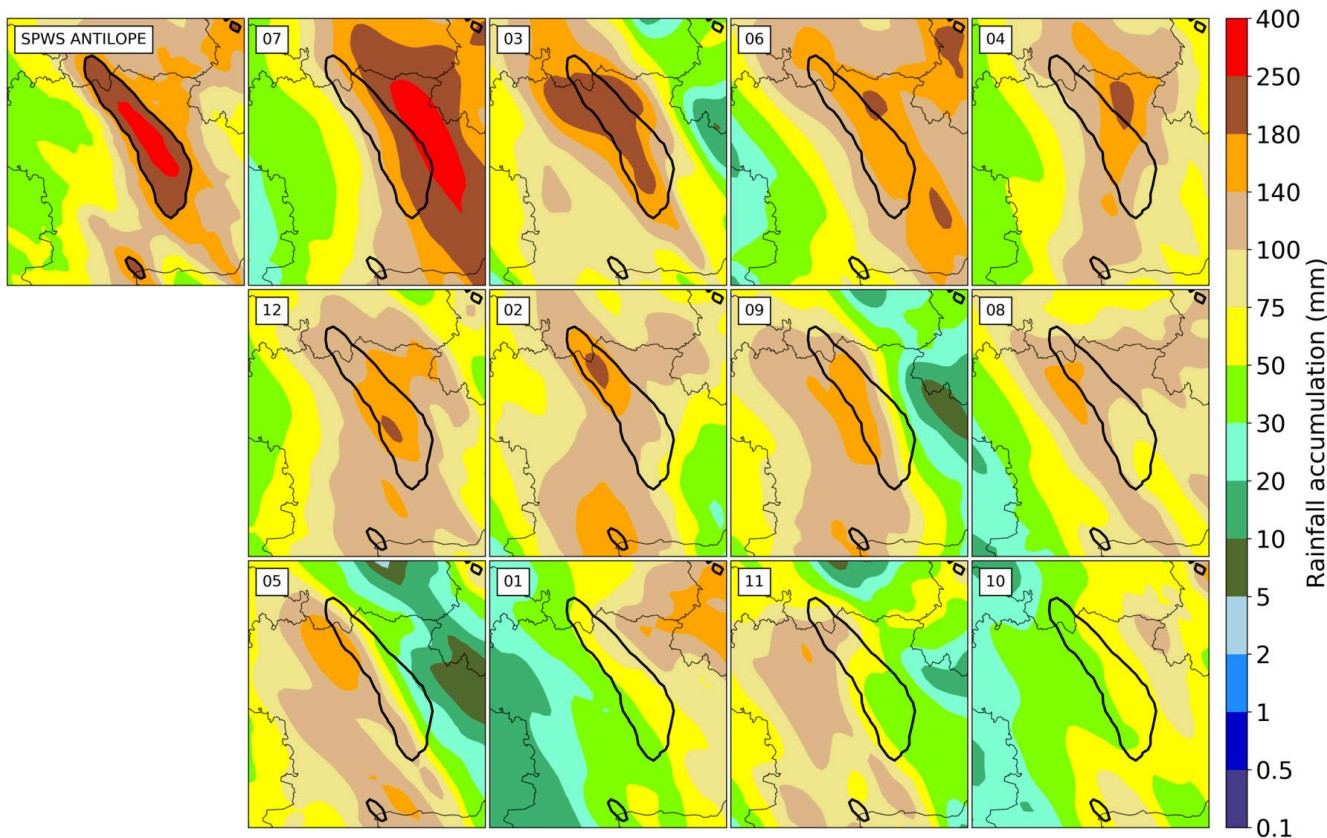

**Figure 16.** Rainfall accumulation over the period of interest observed by SPWS ANTILOPE and predicted by the 12 members of the AROME-EPS run starting at 15:00 UTC on 14 October. Members are ranked by decreasing FSS (from left to right and from top to bottom). The black contour delineates areas where SPWS ANTILOPE observed rain exceeds 180 mm.

high impact weather events. Here, we only have 12 members, so it is not possible to claim that the observed correlations are statistically significant. Focusing on larger-scale features of the anomaly maps, we only attempt to identify clues about the connection between extreme precipitation and other weather fields in this particular event.

Figure 17 indicates that higher precipitation is associated with lower pressure over the Mediterranean Sea, which can be 305 interpreted as a deepening of the mesoscale trough. A deeper trough is expected to be linked with large-scale ascent, which is known to encourage condensation and precipitation. One could expect that a deeper trough would also be linked with stronger wind, but this was not clear on the wind correlation maps (not shown). The correlation maps for both moist potential temperature and relative humidity indicate that higher precipitation is correlated with warmer, moister air over the Mediterranean i.e. upstream of the south-eastern airflow that drove the main precipitating cells. This is consistent with established conceptual 310 models that identify an incident onshore flux of warm, moist air as the key ingredients for Mediterranean heavy precipitation events (e.g. Bresson et al., 2012). Interestingly, our maps also show heavier precipitation to be linked with a colder air mass




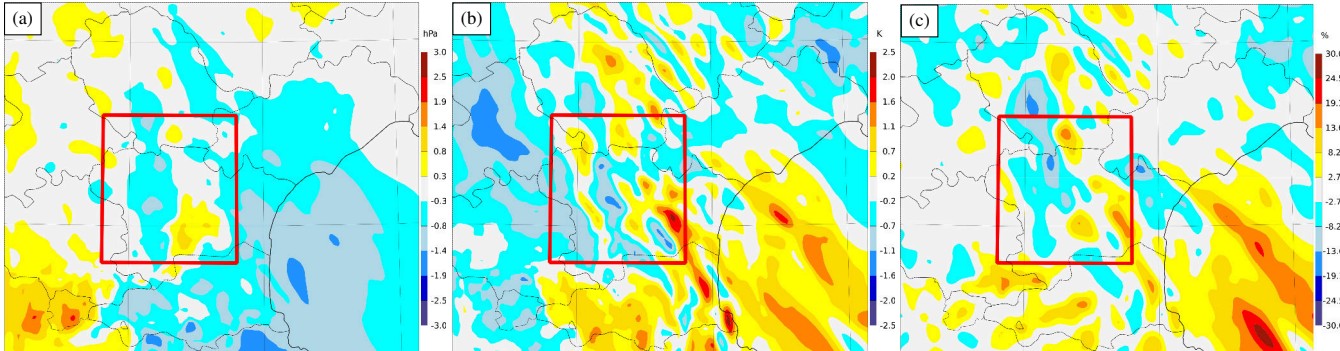

**Figure 17.** Average difference of the three most rainy AROME-EPS members based on 15:00 UTC analysis on 14 October. The forecast time is in the middle of the period. The anomaly is computed with respect to the ensemble mean at each point. From left to right: (a) mean sea level pressure (hPa), (b) moist potential temperature $\theta'_w$ (K) at 850 hPa, and (c) relative humidity (%) at 850 hPa.

near the west of the domain, which confirms the notion that the interaction of the Mediterranean flux with an Atlantic flux was an important mechanism for heavy precipitation in this particular case.

In the following, the focus is on the surface pressure low and the marine low-level jet that are associated with heavy precip-
itation.

### 5.1 Large-scale MSLP low and mesoscale trough observed over the Aude region

At 18:00 UTC, two MSLP low-pressure areas on both sides of the Pyrenees and a high-pressure area over the Alps are observed (fig. 18a). North of the Pyrenees, the MSLP rapidly rose between 18:00 and 20:00 UTC and filled the low-pressure area. To investigate the role of the synoptic low located between Spain and the Balearic Islands during this HPE, its characteristics
are tracked in AROME-France hourly analyses. The low has a non-symmetric shape, and between 18:00 and 22:00 UTC (fig. 18a–b), the development of a trough from the low-pressure area towards the Aude region is observed.

Figure 18a–e show that the centre of the low moved slowly during the entire period. It moved slowly north-north-eastwards between 18:00 and 20:00 UTC, northwards between 20:00 and 22:00 UTC (fig. 18a,b). It remained quasi-stationary between 22:00 and 03:00 UTC (fig. 18b,c), and then moved slowly east-north-eastwards between 03:00 and 06:00 UTC (fig. 18c,d).

In MSLP tendency analyses (fig. 18f–j), we look at the dipoles of pressure decrease-increase that are associated with the movement of the low. Because the low is not symmetric, these dipoles are most of the time also not symmetric. When the low deepens, the decrease part of the dipole predominates and the figure appears globally blue; when the low fills itself, the increased part of the dipole predominates and the figure appears globally red.

Between 21:00 and 00:00 UTC, the low rapidly deepened as shown by MSLP tendencies (fig. 18g). Between 00:00 and
04:00 UTC (fig. 18h), the central pressure of the low remained almost constant, which results in a quite symmetric dipole. The positive part of the pressure tendency dipole associated with the slow movement of the low reached the Pyrenees-Orientales department between 03:00 and 04:00 UTC, stopping the continuous pressure decrease observed from 20:00 until 03:00 UTC


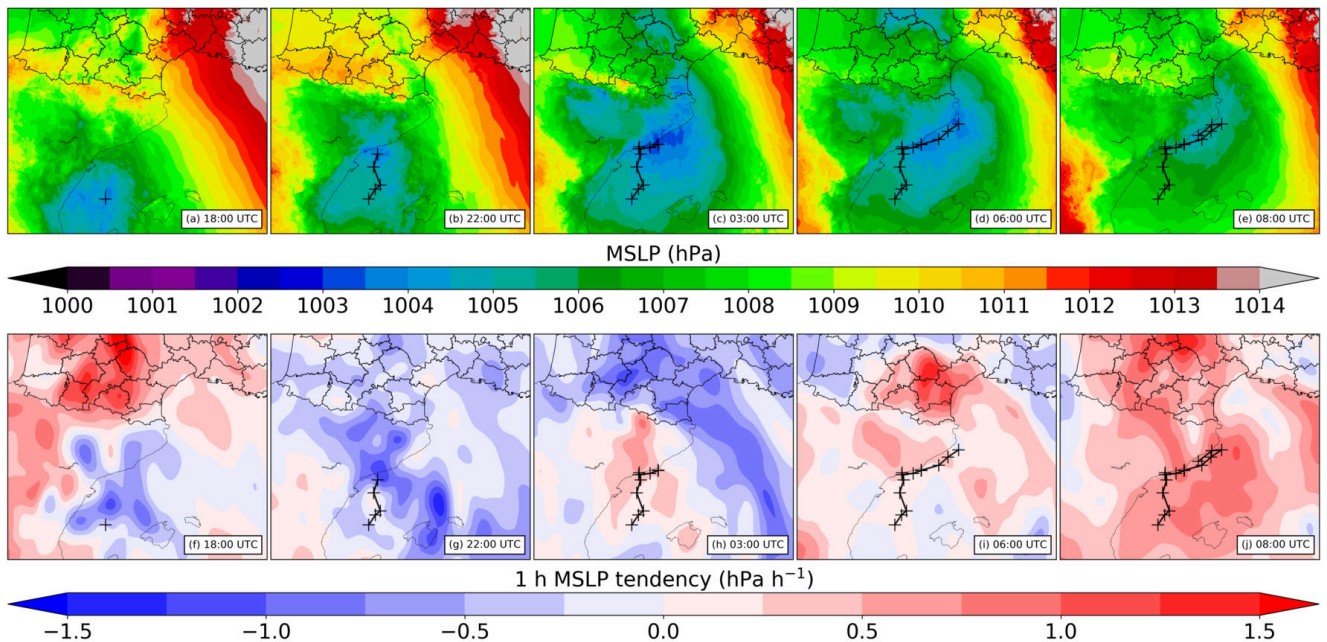

**Figure 18.** AROME-France analyses of (a–e) MSLP and (f–j) 1 h MSLP tendency filtered at 18:00, 22:00 UTC on 14 October and 03:00, 06:00, 08:00 UTC on 15 October.

over the area of interest (fig. 18g,h). This modified the structure of the MSLP field over this area after 04:00 UTC, in particular the location of the trough (fig. 18c–e). After 04:00 UTC (fig. 18i,j), a global pressure rise is observed showing the filling of the 335    low.

There are noticeable differences between AROME-EPS members regarding the spatial structure of the low and the minimum pressure inside the trough over the Aude region (not shown). Because the low-pressure area is not symmetric in all the members, there is no direct relationship between the location of the center of the low and the MSLP over our area of interest. But the spatial structure of the low directly influences the pressure gradient along the Mediterranean shore. In the following section, 340    the relationship of the MSLP gradient delineating the low with the marine low-level wind is investigated.

### 5.2 Marine low-level jet

Time series of departures in meteorological parameters between Sète and Cap Béar (see locations in fig. 19a) are plotted in fig. 19. Figure 19d shows that the MSLP departure between Sète and Cap Béar, along the Mediterranean shore, increased between 18:00 and 23:00 UTC. At the same time, an increase in the mean wind speed in Leucate was observed around 345    19:00 UTC (fig. 19b). Between 21:00 and 04:00 UTC, the MSLP difference between Sète and Cap Béar remained nearly constant between 4.2 hPa and 5.2 hPa. Near the shore, the mean wind speed in Cap Béar remained nearly constant around 21 m s$^{-1}$ in speed and a 140° direction between 22:00 and 04:00 UTC (not shown), while the same constant wind speed is




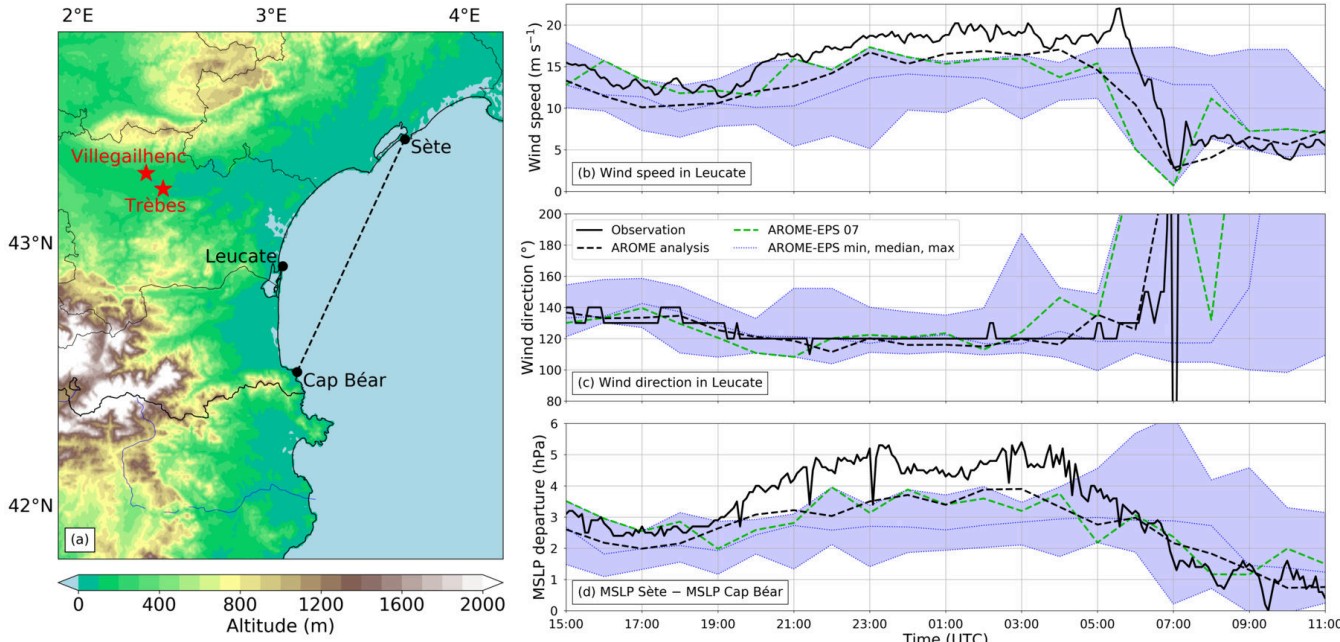

**Figure 19.** (a) Map showing the locations of Sète, Cap Béar and Leucate relative to the region of interest. (b–d) SWS observations, hourly AROME-France analyses and AROME-EPS forecasts of (b) 10 m wind speed in Leucate (c) 10 m wind direction in Leucate and (d) MSLP departure along the Mediterranean shore between 15:00 UTC on 14 October and 11:00 UTC on 15 October. Wind observations are a 10 min mean while models and analyses provide the instantaneous wind at a given time.

observed around 18 m s⁻¹ in speed and a 120° direction in Leucate between 23:00 and 05:00 UTC (fig. 19b,c). Inland, a slow increase of the wind is observed in Lézignan-Corbières from 8 m s⁻¹ to around 13 m s⁻¹, also with a remarkable constant
direction around 110° between 20:00 and 04:00 UTC. The constant direction of the low-level wind observed near the shore and inland shows that even if the MSLP inside the trough decreased during the period, the location and spatial structure of the MSLP field did not evolve much between 20:00 and 04:00 UTC.

AROME-France analyses reproduce well the wind directions observed by all SWSs. The wind decrease and direction shift in Leucate begin approximately one hour earlier in analyses compared to observations. Wind speed in Leucate is higher in
observations than in analyses. Part of this departure is expected because the anemometer in Leucate is located at the top of a 40 m height cliff facing the Mediterranean Sea. Wind speed is better reproduced by the analyses inland over flat terrain. However, the MSLP departure along the Mediterranean Sea is underestimated by 2 hPa in analyses compared to observations.

Figure 19b illustrates that the AROME-EPS median wind speed in Leucate is underestimated by 1.5 to 4 m s⁻¹ between 20:00 and 04:00 UTC compared to the AROME-France analysis and overestimated by 4 to 10 m s⁻¹ between 06:00 and 08:00 UTC.
Moreover, between 01:00 and 04:00 UTC, the AROME-France analysis wind speed is higher than all members of the ensemble. The strong change in wind speed and direction observed between 05:30 and 07:00 UTC is forecast by the ensemble median





between 08:00 and 09:00 UTC. The underestimation of wind speed is correlated with underestimations of the pressure departure along the Mediterranean shore (exactly between Sète and Cap Béar, almost perpendicularly to the isobars). The median MSLP departure forecast is around 2.8 hPa between 21:00 and 04:00 UTC whereas the MSLP departure observation is around 4.7 hPa
during the same time period.

The member of the ensemble with the highest FSS (member 7, also the rainiest) is shown in green. It predicted a wind direction close to the observed one, and it is one of the most windy members between 21:00 UTC and 05:00 UTC as well as one exhibiting the highest MSLP departure between Sète and Cap Béar, above the median of the members during the whole time period.

Figure 20 shows that observations exhibit MSLP departures and mean wind speeds that are higher than all AROME-EPS member forecasts and higher than the AROME-France analysis. The three rainiest members of the ensemble (members 7, 3 and 6) exhibit the closest maritime mean wind speed in Leucate to both observation and AROME-France analysis among the ensemble. This figure illustrates the linear relationship between the MSLP gradient along the shore and the maritime flux.

### 5.3    Stationarity of near-ground mesoscale features

SPWS analyses show that at 18:00 UTC a low MSLP area is observed over the west of the Aude department. Between 19:00 and 21:00 UTC, the MSLP decreases south of the domain shown in fig. 21a, forming the trough shown previously by AROME-France analyses. Around 20:00 UTC, precipitation began and the trough shifted eastwards (fig. 21b). Then, associated with the large-scale MSLP decrease seen in AROME-France analyses, related to the low, a general decrease of MSLP in most of the Aude region is observed from 22:00 UTC until around 04:00 UTC (fig. 21g,h). The axis of the trough remained almost
stationary from 22:00 to 04:00 UTC while it deepened (fig. 22a). It is remarkable, and consistent with MSLP observations, that in most of the domain, except in the south of the Tarn department and under the convective cells, mean wind speed and direction measured by SWSs remained almost stationary between 6 to 8 hours in a row (fig. 19b,c). Then, after 04:00 UTC, MSLP increased as seen in AROME-France analyses (fig. 21i,j). The axis of the trough moved again slowly eastwards, as well as the convective cells (fig. 21d,e). Several SWSs observed the rotation of the wind associated with this movement of the
trough.

Even if AROME-France analyses are most of the time close to SPWS analyses regarding the position of the trough, they underestimate the small-scale signal of stationarity of the trough east of Trèbes seen in SPWS analyses (fig. 22).

From 20:00 UTC on 14 October until the trough vanished around 09:00 UTC on 15 October, the main convective cells of the convective lines remained located near or slightly west of the axis of the trough, resulting in heavy rain slightly west of the
trough (fig. 23). After 04:00 UTC on 15 October, when the axis of the trough moved east towards the Hérault department, this situation persisted. Rainfall was still intense but was not stationary over the same area.

An east-west gradient in 2 m temperature has been observed since the beginning of the event over the Aude region associated to a decaying cold front. Temperatures from 14 to 16 °C are observed west of the Aude department whereas temperatures from 19 to 20 °C east of it, near the Mediterranean Sea. Between 20:00 UTC and 22:30 UTC, the strong temperature gradient moved
slightly from the west of Carcassonne to the east of it, this temperature gradient separating a cold air mass west of the cold





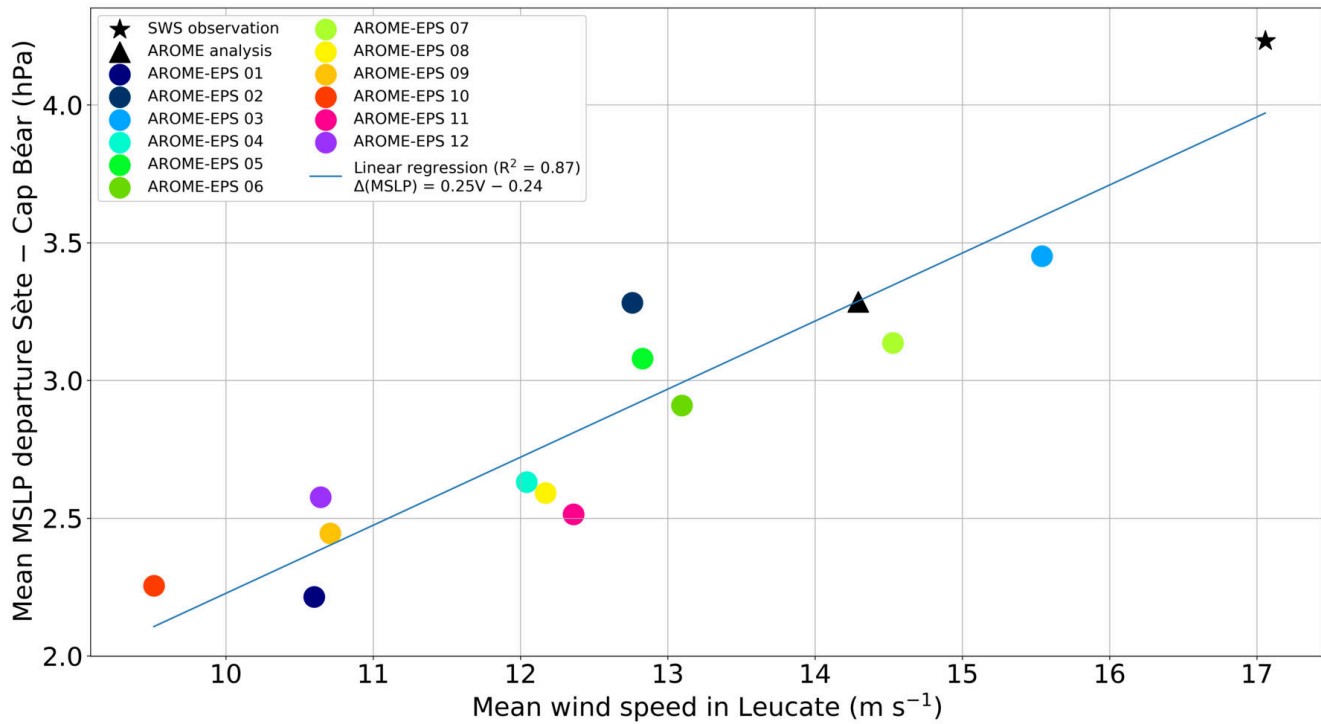

**Figure 20.** SWS observation, AROME-France analysis and AROME-EPS forecasts of 10 m wind speed blowing from the Mediterranean Sea (sector [90°–150°]) as a function of the MSLP departure between Sète and Cap Béar, averaged over the event between 19:00 UTC on 14 October and 07:00 UTC on 15 October (period taken one hour ahead of the period of interest). To give an order of magnitude, if we hypothetically consider that the geostrophic balance applies near the ground, the ageostrophic wind is equal to zero, and the pressure field is such that the Sète – Cap Béar orientation is always parallel to the pressure gradient during the event, the slope of the regression line would theoretically be about 0.13 (with a y-intercept equal to zero).

front from a warm air mass advected from the Mediterranean Sea at the east. Then, between 22:30 UTC on 14 October and 04:00 UTC on 15 October, the strong temperature gradient remained remarkably quasi-stationary. During this period, west of this front, under the convective cells, temperature decrease reached 1.4 °C (e.g. from 14.6 to 13.2 °C at Carcassonne) while near the Mediterranean Sea it remained constant (e.g. from 19.6 to 19.5 °C in Leucate). During the whole event, relative humidity 400 remained high over the region of interest with values ranging from 80 to 100 %. It exceeded 90 % in most of the Aude region after 22:00 UTC. The remarkable stationarity is well captured in virtual potential temperature, a parameter which combines temperature and humidity, from 22:30 to 04:00 UTC (fig. 24a).

The stationarity is remarkable in the Aude valley around 2.5°E (fig. 25), several kilometers east of Villegailhenc and Trèbes, two of the main towns affected by the event. The amplitude of the virtual potential temperature gradient reaches 3 to 4 °C on 405 average over the Aude valley over a distance of 0.2° in longitude (around 16 km). As for the position of the trough, even if AROME-France analyses are most of the time close to SPWS analyses regarding the position of the virtual potential tempera-

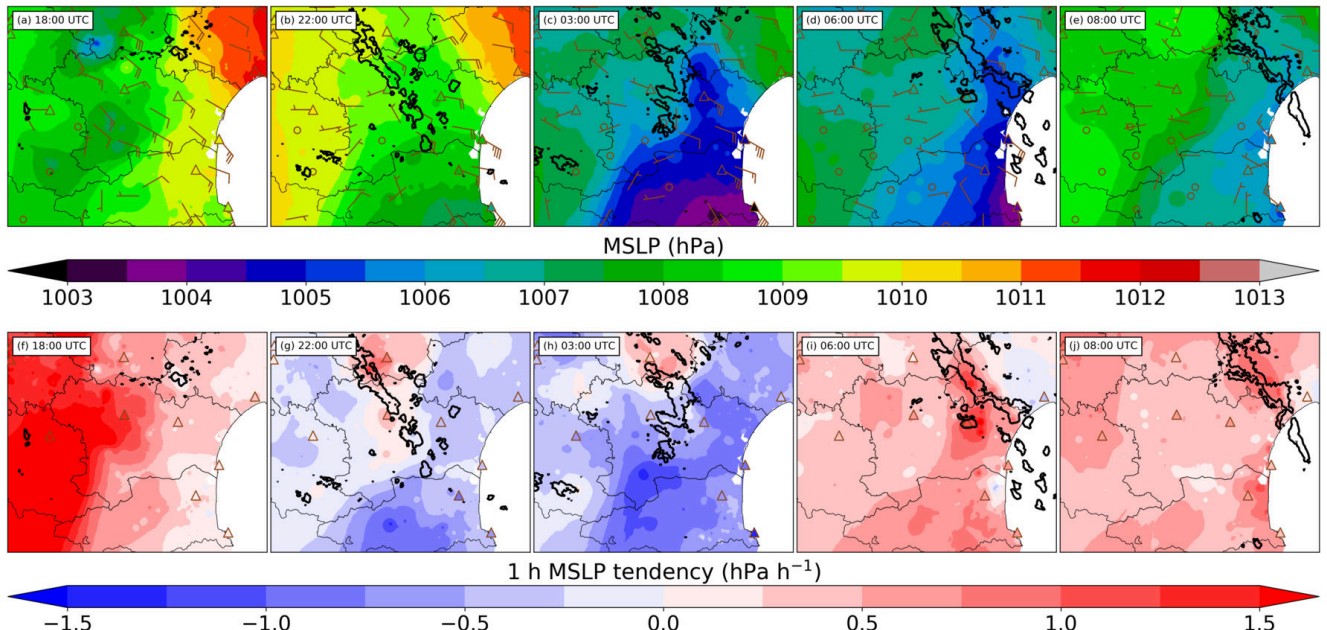

**Figure 21.** SPWS analyses of (a–e) MSLP and (f–j) 1 h MSLP tendency at 18:00, 22:00 UTC on 14 October and 03:00, 06:00, 08:00 UTC on 15 October. Brown triangles show SWS MSLP observations, and brown barbs show SWS wind observations. Bold black contours indicate radar reflectivities above 40 dBZ.

ture gradient (fig. 24b), they underestimate the small-scale signal of stationarity of this gradient east of Trèbes seen in SPWS analyses.

By combining SWSs with PWSs, we identified two main near-surface features correlated with the stationarity of the heavy-
precipitation system:

  – a quasi-stationary MSLP trough around 0.25° east in longitude of the heavy precipitation,

  – a quasi-stationary virtual potential temperature gradient 0.1° east in longitude of the heavy precipitation.

The virtual potential temperature gradient materializes the boundary between the cold air located at the west of the region, and the warm maritime air advected from the Mediterranean Sea. During the period of stationarity of rainfall, as the 2 m
temperature decreases under the convective system, we could hypothesise that it is due to evaporation processes. It has already been shown that slow-moving cold fronts or cold pools play a major impact on the location of heavy precipitating events in the Mediterranean region (Ducrocq et al., 2008; Fiori et al., 2017; Duffourg et al., 2018). In this case, near the ground, observations show an existing cold air mass before the event begins, which is cooled during the event, increasing the west-east gradient of temperature along the front.
In addition to this boundary, the slow-moving synoptic low extended by a mesoscale quasi-stationary trough probably helped to enhance near-surface convergence, and subsequently focus the convection over the Trèbes area. The presence of mesoscale



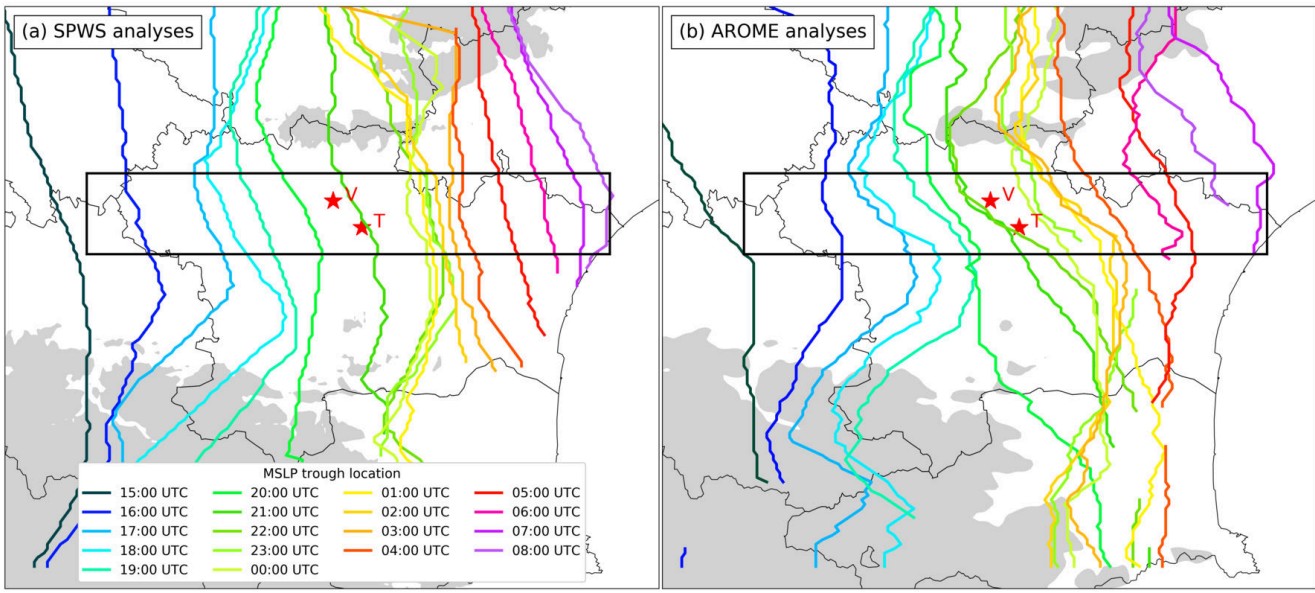

**Figure 22.** Location of the MSLP trough as a function of time between 15:00 UTC on 14 October and 08:00 UTC on 15 October indicated by (a) SPWS analyses and (b) AROME-France analyses. The Aude valley domain is indicated by the black box. Stars annotated "T" and "V" show the location of Trèbes and Villegailhenc, respectively, two towns severely damaged. Terrain above 750 m a.s.l. is shaded in grey.

surface disturbances such as a trough or a mesolow, modifying the circulation near the ground during Mediterranean HPE has already been noted by Romero et al. (2000); Nuissier et al. (2008); Ducrocq et al. (2008).

There may be some similarities between the HPE of November 1999 also in the Aude region and the 2018 case: the presence

of a surface low in ALADIN forecasts was noted in the vicinity of the system and the correlation between the strong radar echoes and the area of strong dew point temperature gradient was also remarked (Aullo et al., 2002).

Figure 25 shows that the exact location of the precipitation predicted by the AROME-EPS system over the Aude valley is strongly correlated with the location of the predicted virtual potential temperature gradient, located east of the precipitation, as in the observations. Also, the highest hourly rain rates are associated with strong virtual potential temperature gradients (> 3 °C

difference in less than 0.2° of longitude). If we focus on the three rainiest members (7, 3, and 6), they show strong gradients of virtual potential temperature in the Aude valley region, located east of the strongest rainfall, as seen in observations. Member 7 shows a quasi-stationary virtual potential temperature gradient near 2.65°E during 6 hours, close to the stationarity seen in observations around 2.5°E also during almost 6 hours. Even if the location of the gradient is shifted eastwards, its evolution from west to east is very similar to the observed one. The temperature gradient is a little stronger than observed. Members 3

and 6 show temperature gradients and trough oscillations during the case but with no clear stationarity as seen in member 7. These oscillations result in too large predicted rainfall areas and too low peak rainfall accumulations.


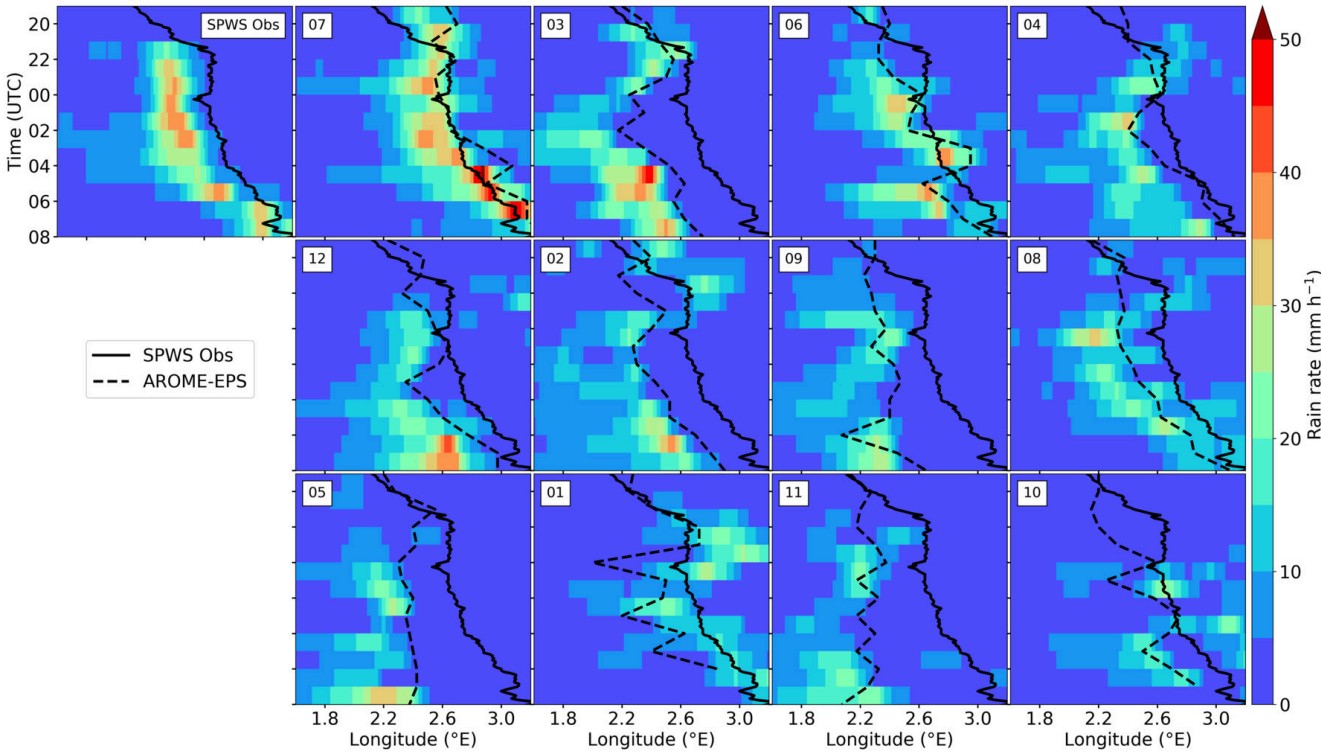

**Figure 23.** Hovmöller diagram of the mean trough's longitude over the Aude valley observed with SPWS analyses and predicted by the members of the AROME-EPS between 19:00 UTC on 14 October and 08:00 UTC on 15 October. The mean hourly rain rate observed by SPWS ANTILOPE or predicted by AROME-EPS over the Aude valley [43.15°N–43.33°N] is shown as a function of the longitude. Members are ranked as in fig. 16.

Figure 23 shows that in most of the members (all except 1 and 10), the location of rainfall is also correlated with the location of the trough. The reason why it is not correlated in members 1 and 10 is probably because the trough is almost non-existing over the area in these members, which leads to identify locations that are isolated local MSLP minima.

**6  Conclusions**

The case of the deadly flash floods in the Aude catchment area on 14 and 15 October 2018 was studied on several scales using operational numerical weather prediction systems and observations including observations of connected objects. On a large scale, the meteorological situation is characterised by high geopotential values at 500 hPa over central Europe and slowly evolving geopotential anomalies over the western Atlantic and the Iberian Peninsula. These are classically found in 445 Mediterranean HPEs, especially in the one of 12 and 13 November 1999 which caused flash floods in the same region (Nuissier et al., 2008).

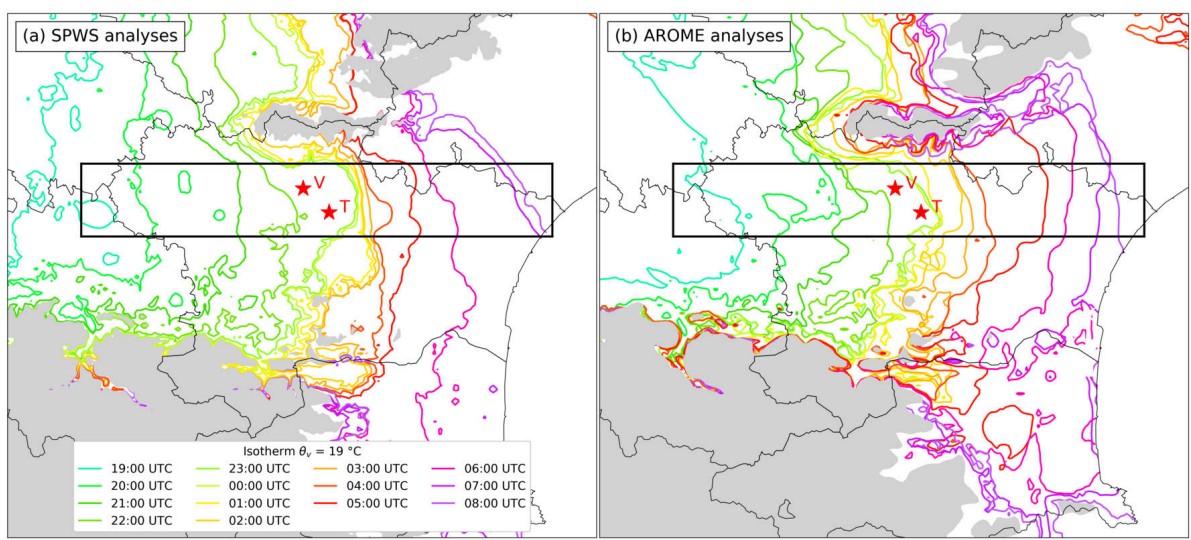

**Figure 24.** As in fig. 22 showing the location of the 19 °C virtual potential temperature isotherm, starting at 19:00 UTC 14 October.

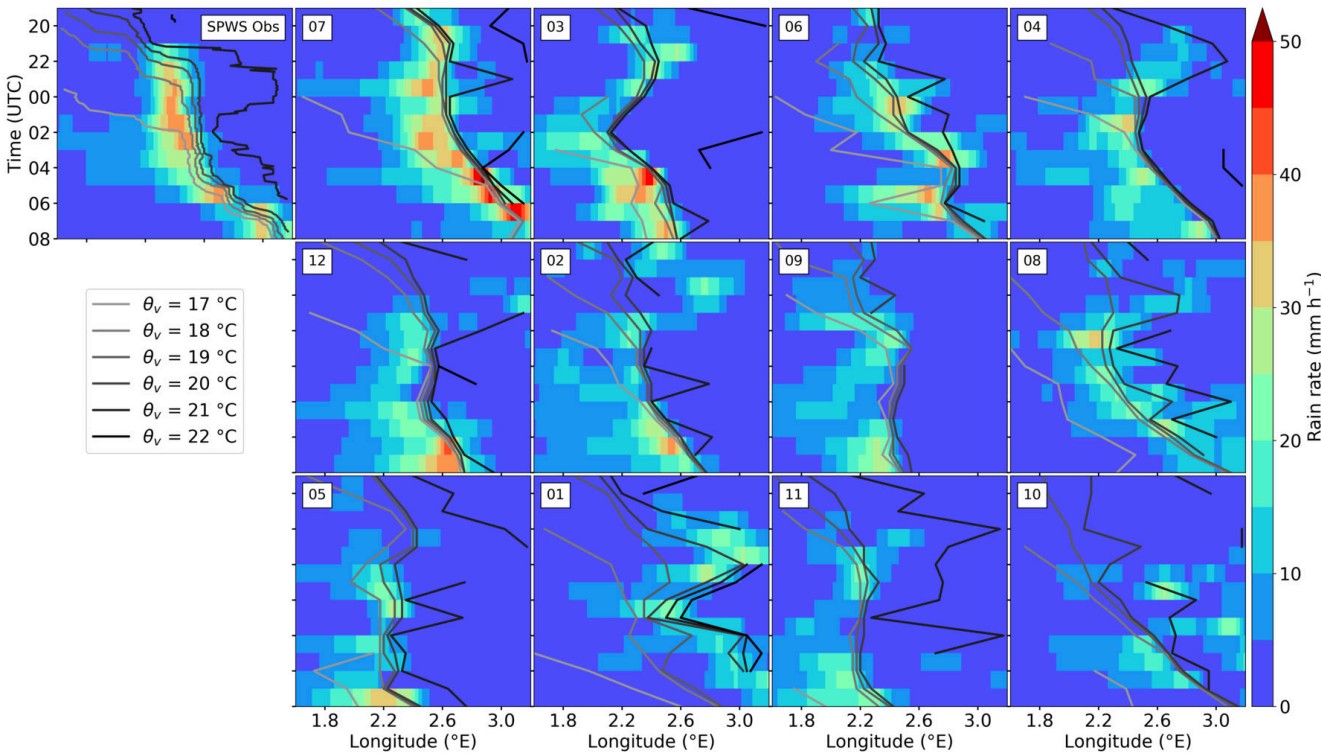

**Figure 25.** As in fig. 23 showing the mean longitude of the 17, 18, 19, 20, 21 and 22 °C virtual potential temperature isotherms.





Less classic was the presence of the remnants of a hurricane that may have brought additional moisture. Our study did not reveal any particular role of the former hurricane Leslie in the contribution of humidity to the convective system that flooded the Aude basin. To evidence such a contribution, it would be necessary, for example, to carry out numerical simulations with backward trajectories to determine the origin of the humidity. Another potential source of humidity is traditionally the Mediterranean Sea. It has been shown that strong evaporation has taken place in the western part of the Mediterranean basin, which most likely contributed to the supply of moisture to the convective system. Once again, backward trajectories could make it possible to quantify this effect.

Examination of the radar reflectivities revealed the presence of a continuous train of convective cells starting from the eastern tip of the Pyrenees mountain range and heading towards the Aude catchment. The eastern end of the Pyrenees seems to play a role in the initiation of convection, which a numerical study of the sensitivity to the presence of the relief could confirm.

From a hydrological point of view, it has been shown that soil moisture did not play a significant role in the formation of floods in the Aude basin. Predictability of basin rainfall with a set of NWP systems (deterministic, probabilistic, oriented towards nowcasting) has been shown to be limited despite predicted rainfall that was often realistic but offset with respect to the catchments where the highest rainfall was observed—a well-known phenomenon in the field of hydrometeorological forecasting of flash floods (e.g., Hally et al., 2015).

The analysis of the link between heavy rainfall and other fields predicted by AROME-EPS showed particular mesoscale meteorological patterns in the rainiest members. The presence of a mesoscale trough near the ground was evidenced in the area of interest, as well as a moist, warm low-level jet coming from the Mediterranean and colder air west of the heavy precipitation. The mesoscale trough connected to a low located between Spain and the Balearic Islands which moved slowly north-eastwards during the event. The low deepened until 00:00 UTC on 15 October, between 00:00 and 04:00 UTC, MSLP remained quite constant and then the low started filling up after 04:00 UTC. It was shown that the marine low-level jet wind speed is linearly related to the pressure gradient around the low. AROME-France analyses and the rainiest AROME-EPS members exhibit the largest pressure gradients and low-level jet wind speeds, even if they do not reach the values observed by surface stations.

The use of observations from personal weather stations in addition to standard stations has improved precipitation estimates and helped give a picture of the state of the atmosphere near the ground with unprecedented detail. With supplementary surface observations, small-scale near-surface features correlated to the stationarity of the rainfall observed were brought to light.

The mesoscale trough was found to remain quasi-stationary east of the largest rainfall accumulations in the Aude valley during approximately 6 hours. The observed deepening of the trough may have been amplified by intense lifting, which in turn helped increase the near-surface marine flux and convergence. Cold air was advected from the Atlantic on 14 October by a cold front during the day. The remnants of the cold front formed a pre-existing west-east virtual potential temperature gradient, and cold air was observed moving from the western part of the Aude towards east before the event. The gradient strengthened rapidly from 22:30 UTC and remained almost stationary just after rainfall began. It was located during the whole event east of the largest rainfall accumulations. This cold boundary may have caused enhanced lifting of air parcels, helping to focus thunderstorms over the area where the maximal rainfall was observed. Moreover, the cooling associated with rainfall evaporation, by increasing the temperature gradient, may have provided a positive feedback to lifting enhancement.



The stationarity of these two features between 22:30 and 04:00 UTC showed that a quasi-equilibrium between the near-surface marine flux feeding the system from the Mediterranean Sea and the expansion of the cold boundary towards east was reached. This equilibrium seems to have lasted until the low over the Mediterranean Sea moved north-eastwards and caused a
pressure increase in the Aude region starting from 04:00 UTC, shifting the trough and the temperature gradient as well as the thunderstorms from the stationarity area.

Operational AROME-EPS members globally underestimated the rainfall over the Aude valley. Only the rainiest member of the ensemble was able to forecast a rainfall amount exceeding 320 mm over the Aude valley, approximately 20 km east of the area exceeding 300 mm in the observations. The other members generally located the maximum rainfall over the mountains.
Regarding the near-surface features, significant differences in the MSLP low amplitude and trajectory are seen among members. They affect in turn the forecasts of the amplitude and location of the trough associated with it as well as the predicted maritime flux.

The three rainiest members predicted the three strongest mean wind speeds blowing from the Mediterranean Sea in Leucate. Among these three members, only the rainiest member of the AROME-EPS reproduced quasi-stationary MSLP trough and
virtual potential temperature gradient similar to the observations.

To investigate the initiation processes, the impact of the Pyrenees relief, the role of the diabatic cooling, the origin of moisture, and the role of changes in wind near the Mediterranean shore in the stationarity of this backbuilding multicellular system, further storm-scale numerical simulations will be run in a future study.

*Code and data availability.* The codes of the operational NWP systems mentioned in the manuscript are not free, but the forecasts can
be obtained upon request from the authors for research purposes. The OSTIA SST fields were provided by GHRSST, Met Office and CMEMS (METOFFICE-GLO-SST-L4-NRT-OBS-ANOM-V2 product on http://marine.copernicus.eu/). The hydrological observational data are available from the French HYDRO data bank (http://www.hydro.eaufrance.fr/, last access: 19 August 2020) or come from post-event surveys conducted in the framework of HyMeX (data set available here: https://mistrals.sedoo.fr/?editDatsId=1512&datsId=1512&project_name=HyMeX&q=post-event+survey). The weather observations can be obtained upon request from the authors for research purposes.

*Author contributions.* All authors collaborated and contributed to drafting, reviewing, and editing the paper. In particular, OC coordinated
the writing of the paper; MM contributed to the analysis of radar-raingauge data and to the analysis of the relationship between rainfall and mesoscale features; FB contributed to the analysis of AROME-EPS forecasts; JE contributed to the analysis of soil moisture and hydrological data; CLB contributed to the analysis of atmosphere-ocean interactions; AL contributed to the analysis of hydrological data and AROME-NWC forecasts. ON contributed to the analysis of the meteorological context; and OL contributed to the analysis of radar-raingauge data.

*Competing interests.* The authors declare that no competing interests are present.





*Acknowledgements.* This work is a contribution to the HyMeX programme supported by MISTRALS and ANR PICS grant ANR-17-CE03-0011. The map tiles in fig. 5 are by Stamen Design, under CC BY 3.0; data by OpenStreetMap, under ODbL.



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
