# Peer review of "The heavy precipitation event of 14–15 October 2018 in the Aude catchment: A meteorological study based on operational numerical weather prediction systems and standard and personal observations"

_Natural Hazards and Earth System Sciences, 2020_

## Referee Comment (RC1) · Anonymous Referee #1 · 20 Oct 2020

The paper by Caumont et al. (1) describes a heavy precipitation event, (2) discusses the inclusion of personal weather stations in precipitation analyses, and (3) tries to evaluate numerical weather forecasts. Consequently, it is an interesting manuscript with many details, but it is a long and sometimes confusing manuscript too. It is overall well written and illustrated. I suggest focusing on one of the three items, and thereby rewriting and shortening the manuscript substantially.

[Figure]

Suppose the focus shall be on event description (1). In that case, I think it will be worth trying to quantify the impact of the mentioned leftovers of a former hurricane, which made the "classic synoptic situation" (line 4) special. These hurricane leftovers were mentioned already in the Abstract, and the reader is hoping for a more in-depth consideration. But, the authors only wrote vaguely in the Conclusion "did not reveal any particular role" and mention that trajectory studies would have been necessary (line 450). They mention (line 452) strong evaporation over the Western Mediterranean Sea, which "most likely contributed to the supply of moisture". It would be interesting to learn if strong winds related to the hurricane leftovers lead to the strong evaporation. The ARPEGE and AROME analysis could help in the discussion of the event processes, but the distraction of discussing the skill of the AROME now- and forecasts should be skipped. Finally, it would be important to discuss the features of this "classic" event in the context of other classic events in the area and beyond in the Mediterranean basin (not France only).

The discussion (2) about the impact of personal weather stations could be kept very short and moved to an Appendix. The discussion of the added value of personal weather stations in the QPE should not be mixed with the event description in Sec. 3.2.

If evaluation (3) of the AROME now- and forecasts shall be the manuscript's goal, this has to be more conclusive. For example, the authors wrote in the Abstract that the rainfall forecasts had limited predictability (line 9). Which forecast? The forecasts of ARPEGE, AROME-FRANCE, AROME-NWC, or all of the AROME-EPS members? In the Abstract and Conclusion, it is mentioned that the best forecast (one of the EPS members) contained the warmest, wettest, and fastest low-level jet. The EPS is introduced in one sentence only (lines 87-88!) without hinting at the applied perturbation method. Was the one good forecast member just luck? Has the EPS any predictive value for events like the discussed one? Why did the other members miss the important mesoscale features?

The discussion of river network, runoff, infiltration, etc. (e.g., page 13) could be skipped or very briefly done in the introduction. The comments on soil moisture (e.g., line 8) and its role in flood formation (line 458) are misleading as the authors neither discuss any precipitation - soil moisture feedback/recycling, nor discuss the flood event in depth.

———————————————

---

## Referee Comment (RC2) · Anonymous Referee #2 · 28 Oct 2020

Review of the manuscript NHESS-2020-310

The heavy precipitation event of 14–15 October 2018 in the Aude catchment: A meteorological study based on operational numerical weather prediction systems and standard and personal observations

submitted for publication to Natural Hazards and Earth System Sciences

[Figure]

The paper presents the analysis of a case of heavy precipitation which led to several damages in south-western France. This case study is analyzed by means of observations and different operational numerical weather prediction models. The manuscript is interesting and the analyses presented are rather extensive. However, in the current form, the presentation of the event and of the results is confusing. Moreover, there are too many figures. I therefore suggest reorganizing the manuscript in order to improve its readability.

Major remarks

1) I would completely remove the hydrological description of the event, since it is not important and essential for the other parts of the paper. Therefore, I would remove lines 197-216, Figs. 8 and 9 and Tables 1 and 2.

2) In my opinion Section 5 is not easy to follow. There is a mix between the mesoscale description of the event and the evaluation and interpretation of numerical weather prediction models. I would move all the parts where the event is simply described to Section 3, where the case study can be described both at the synoptic scale and at the mesoscale. So, for example, I would move the description contained in Section 5.1 to Section 3.

3) Figure 17: Authors say: "Figure 17 indicates that higher precipitation is associated with lower pressure over the Mediterranean Sea, which can be interpreted as a deepening of the mesoscale trough". The lower pressure highlighted in Fig. 17a may be caused by a different timing of the low pressure moving eastward. In my opinion, it would be clearer showing the pressure field of the three members with higher precipitation (all the members or the average) and of the ensemble mean. In this way, the position and the values of the low pressure can be directly appreciated and compared. I think that this comparison would be useful also in the following section where the intensity of the low-level wind is analyzed. The same considerations apply also for panel b). In panel c) I would plot the specific humidity and not the relative humidity, to have

an idea of the absolute amount of water vapor. Finally, it is not clear what time is "the middle of the period".

4) In Section 5.3 the Authors discuss the relation between the virtual potential temperature gradients and the localization of heavy precipitations. In my opinion, this temperature gradient is a consequence of the precipitations, so it is not so important for the analysis of the different performance of the members of the ensemble. The Author say: "In this case, near the ground, observations show an existing cold air mass before the event begins, which is cooled during the event, increasing the west-east gradient of temperature along the front". However, this is not shown in the paper, so it is not clear how this cold air mass may have influenced the localization of the precipitations.

Minor remarks

Abstract: the Authors say: "it is shown that the positive Mediterranean sea surface temperature anomaly may have played an aggravating role in the amount of precipitation...". However, this is just a speculation of the Authors, which is not demonstrated in the paper. Therefore, I suggest removing this sentence from the abstract.

Page 2, line 55: I do not understand why this event is "atypical".

Figure 3: I suggest keeping only one of the four panels of Fig. 3, as the information contained in the four panels is very similar.

---

## Author Comment (AC1) · 22 Jan 2021

*We thank Anonymous Referee #1 for his/her constructive comments. Our reply is in blue and quotes from the revised manuscript are in purple. Line numbers correspond to the original submission.*

**Anonymous Referee #1**

The paper by Caumont et al. (1) describes a heavy precipitation event, (2) discusses the inclusion of personal weather stations in precipitation analyses, and (3) tries to evaluate numerical weather forecasts. Consequently, it is an interesting manuscript with many details, but it is a long and sometimes confusing manuscript too. It is overall well written and illustrated. I suggest focusing on one of the three items, and thereby rewriting and shortening the manuscript substantially.

> The manuscript now focuses on the use of operational weather forecasts and standard and crowd-sourced observations to highlight the meteorological processes that characterise this extreme hydrometeorological event. The text has been significantly shortened. The text is now 32 pages long instead of 38 initially. For instance, the part discussing the inclusion of personal weather stations in precipitation analyses—which we believe to be interesting *per se*, but not essential in the presentation of the article's main objective—has been moved to an appendix. The evaluation of the numerical weather forecasts has been reduced to what is strictly necessary to validate their use in Section 5. Thus, the evaluation of AROME-NWC has been deleted.

Suppose the focus shall be on event description (1). In that case, I think it will be worth trying to quantify the impact of the mentioned leftovers of a former hurricane, which made the "classic synoptic situation" (line 4) special. These hurricane leftovers were mentioned already in the Abstract, and the reader is hoping for a more in-depth consideration. But, the authors only wrote vaguely in the Conclusion "did not reveal any particular role" and mention that trajectory studies would have been necessary (line 450). They mention (line 452) strong evaporation over the Western Mediterranean Sea, which "most likely contributed to the supply of moisture". It would be interesting to learn if strong winds related to the hurricane leftovers lead to the strong evaporation. The ARPEGE and AROME analysis could help in the discussion of the event processes, but the distraction of discussing the skill of the AROME now- and forecasts should be skipped. Finally, it would be important to discuss the features of this "classic" event in the context of other classic events in the area and beyond in the Mediterranean basin (not France only).

> We understand the reviewer's expectations regarding the role of former hurricane Leslie. The outcome of our work is that our tools are not able to draw any definite conclusion about the role of Leslie and therefore calls for further investigations with more appropriate tools such as research atmospheric models with the capability of tracing the origin of moisture back. The respective roles of SST and low-level wind velocity on evaporation could be quantified in models thanks to Spearman's rank correlations as done for example in Bouin and Lebeaupin Brossier (2020), but are more relevant if applied on ocean-atmosphere coupled numerical systems. Such a coupled system is currently under development and we clearly identify this situation as a golden case for further investigating the sea upper layer role on moisture feeding.
> As proposed by the reviewer, the distraction of discussing the skill of the AROME now- and forecasts has been mostly skipped and has been reduced to the minimum necessary to validate our tools on this case. More references to other heavy precipitation events in the Mediterranean basin were added.

The discussion (2) about the impact of personal weather stations could be kept very short and moved to an Appendix. The discussion of the added value of personal weather stations in the QPE should not be mixed with the event description in Sec.3.2.

> The discussion about the impact of personal weather stations has been moved to Appendix A.

If evaluation (3) of the AROME now- and forecasts shall be the manuscript's goal, this has to be more conclusive. For example, the authors wrote in the Abstract that the rainfall forecasts had limited predictability (line 9). Which forecast? The forecasts of ARPEGE, AROME-FRANCE, AROME-NWC, or all of the AROME-EPS members? In the Abstract and Conclusion, it is mentioned that the best forecast (one of the EPS members) contained the warmest, wettest, and fastest low-level jet. The EPS is introduced in one sentence only (lines 87-88!) without hinting at the applied perturbation method. Was the one good forecast member just luck? Has the EPS

any predictive value for events like the discussed one? Why did the other members miss the important mesoscale features?

> The evaluation of models is no longer a goal of the article. We have therefore shortened the related text. We kept our (short) conclusions about the predictability of the event because it is a property of the meteorological event at forecast time, not of the numerical forecasts used, so it cannot be ascribed to a particular NWP run. On lines 234–235 it is explained that predictability is estimated using successive AROME-France runs. In order to clarify it, we changed the corresponding sentence which now reads:

> This complicates the work of forecasters and crisis managers, since the spread of successive AROME-France runs indicates that the event had limited predictability.

> So as to further clarify this point, the sentence in lines 280–282 has been changed to:

> In summary, fig. 8 indicates that, although the higher quantiles of the ensembles provided an accurate representation of the overall event in space, time and intensity, the predictability of the event was rather low according to AROME-EPS (as already indicated by AROME-France in section 4.1): prediction could not have been forecast with less than a $50\%$ uncertainty on intensity, and about 3 hours in terms of timing.

> Concerning the description of AROME-EPS, some details have been added and it now reads as follows:

> AROME-EPS is a 12-member ensemble based on perturbations of the AROME-France model at a resolution of $2.5\,\mathrm{km}$ in 2018. The AROME-EPS system is updated every six hours and it samples the forecast uncertainties using perturbations of the initial condition (atmosphere and surface), large-scale coupling, and model equation (using stochastic physics perturbations). The system is extensively documented in the references given above.

> The following text is added at the beginning of section 4.2, which refers to section 5 where a physical interpretation of the AROME-EPS members is provided:

> Although they are not perfect tools, convection-permitting ensembles like AROME-EPS are known to provide valuable information about the probability distribution of Mediterranean heavy precipitation events (see e.g. Hally et al. 2015). In this section we present the forecasted probability distribution. Some discussion of the link between member performance and its physical behaviour is provided in section 5.

The discussion of river network, runoff, infiltration, etc. (e.g., page 13) could be skipped or very briefly done in the introduction. The comments on soil moisture (e.g., line 8) and its role in flood formation (line 458) are misleading as the authors neither discuss any precipitation - soil moisture feedback/recycling, nor discuss the flood event in depth.

> Most of the hydrological part of the article has been skipped. We kept the brief comment regarding the soil wetness because it could have been one of the meteorological factors explaining the intensity of the floods and the main goal of the article is now to highlight the meteorological processes that characterise this extreme hydrometeorological event.

**References**

Bouin, M.-N. and Lebeaupin Brossier, C.: Surface processes in the 7 November 2014 medicane from air-sea coupled high-resolution numerical modelling, Atmospheric Chemistry and Physics, 20, 6861–6881, https://doi.org/10.5194/acp-20-6861-2020, 2020.

---

## Author Comment (AC2) · 22 Jan 2021

*We thank Anonymous Referee #2 for his/her constructive comments. Our reply is in* blue *and quotes from the revised manuscript are in* purple. *Line numbers correspond to the original submission.*

**Anonymous Referee #2**

Review of the manuscript NHESS-2020-310
*The heavy precipitation event of 14–15 October 2018 in the Aude catchment: A meteorological study based on operational numerical weather prediction systems and standard and personal observations*
submitted for publication to Natural Hazards and Earth System Sciences

The paper presents the analysis of a case of heavy precipitation which led to several damages in south-western France. This case study is analyzed by means of observations and different operational numerical weather prediction models. The manuscript is interesting and the analyses presented are rather extensive. However, in the current form, the presentation of the event and of the results is confusing. Moreover, there are too many figures. I therefore suggest reorganizing the manuscript in order to improve its readability.

> The number of figures has been reduced from 25 initially, down to 18. The objective of the paper has been reduced to the use of operational weather forecasts and standard and crowdsourced observations to highlight the meteorological processes that characterise this extreme hydrometeorological event. Numerous details, which were not essential to the body of the manuscript, and passages that are now off-topic have been either left out or sent to an appendix, with the intent of improving the manuscript's readability.

**Major remarks**

1) I would completely remove the hydrological description of the event, since it is not important and essential for the other parts of the paper. Therefore, I would remove lines 197-216, Figs. 8 and 9 and Tables 1 and 2.

> The hydrological description of the event has been significantly shortened. However, we kept the brief comment regarding the soil wetness because it could have been one of the meteorological factors explaining the intensity of the floods and the main goal of the article is now to highlight the meteorological processes that characterise this extreme hydrometeorological event.

2) In my opinion Section 5 is not easy to follow. There is a mix between the mesoscale description of the event and the evaluation and interpretation of numerical weather prediction models. I would move all the parts where the event is simply described to Section 3, where the case study can be described both at the synoptic scale and at the mesoscale. So, for example, I would move the description contained in Section 5.1 to Section 3.

> Section 3 is intended to provide an overview of the event without going into detail. The ARPEGE model analyses are used for this purpose, as one could have used those of IFS for example. Section 5 aims to use both deterministic and ensemble-based fine-scale models based on Arome to explain the meteorological processes involved in the event at mesoscale in more detail.

3) Figure 17: Authors say: "Figure 17 indicates that higher precipitation is associated with lower pressure over the Mediterranean Sea, which can be interpreted as a deepening of the mesoscale trough". The lower pressure highlighted in Fig. 17a may be caused by a different timing of the low pressure moving eastward. In my opinion, it would be clearer showing the pressure field of the three members with higher precipitation (all the members or the average) and of the ensemble mean. In this way, the position and the values of the low pressure can be directly appreciated and compared. I think that this comparison would be useful also in the following section where the intensity of the low-level wind is analyzed. The same considerations apply also for panel b). In panel c) I would plot the specific humidity and not the relative humidity, to have an idea of the absolute amount of water vapor. Finally, it is not clear what time is "the middle of the period".

> We are sorry that limits to the length of the paper do not allow us to present the ensemble members maps for all the interesting parameters. The ensemble mean is not a very informative plot because it tends to smooth out important features of the flow, and to confuse the graphical interpretation with many small-scale details. However, we have investigated the interesting point raised by the reviewer and for this we have looked at individual members. The cited sentence has been replaced with the following one:

Figure 10 indicates that higher precipitation is associated with lower pressure over the Mediterranean Sea, which is both due to a larger and earlier deepening of the mesoscale trough as indicated by a visual examination of the pressure fields of all members (not shown).

The 'middle of the period' referred to the 'middle of the period of interest'. It has been clarified in the caption:

The forecast time is 02:00 UTC on 15 October 2018, i.e., in the middle of the period of interest.

Specific humidity was examined too, but its correlation with precipitation is much less clear, which is understandable because the condensation processes that drive convection and precipitation tend to be driven by the occurrence of saturation, rather than the absolute water content. Saturation being sensitive to both specific humidity and temperature it is more effectively diagnosed by relative humidity than specific humidity.

4) In Section 5.3 the Authors discuss the relation between the virtual potential temperature gradients and the localization of heavy precipitations. In my opinion, this temperature gradient is a consequence of the precipitations, so it is not so important for the analysis of the different performance of the members of the ensemble. The Author say: "In this case, near the ground, observations show an existing cold air mass before the event begins, which is cooled during the event, increasing the west-east gradient of temperature along the front". However, this is not shown in the paper, so it is not clear how this cold air mass may have influenced the localization of the precipitations.

In fact, further investigations, which are the subject of another article following this one, show that the contribution of precipitation to the formation of the temperature gradient is small. It is precisely the work shown in this article that has made it possible to highlight the interest of studying the question of the influence of precipitation on the temperature gradient between the two air masses.

**Minor remarks**

Abstract: the Authors say: "it is shown that the positive Mediterranean sea surface temperature anomaly may have played an aggravating role in the amount of precipitation...". However, this is just a speculation of the Authors, which is not demonstrated in the paper. Therefore, I suggest removing this sentence from the abstract.

The sentence has been rephrased as follows:

Mediterranean Sea surface temperature and soil moisture anomalies are briefly reviewed, as they are known to play a role in this type of hydrometeorological event.

Page 2, line 55: I do not understand why this event is "atypical".

The event is atypical because of the joint presence of a former hurricane (Leslie) and a near-ground cold air mass. This has been clarified in the text by splitting the sentence in two:

This case was chosen because it has had particularly dramatic consequences, but also because of the atypical joint presence of a former hurricane (Leslie) and a near-ground cold air mass. The 2018 floods took place nineteen years after one of the major precipitating episodes recorded in the same region, the episode of 12–13 November 1999 (Nuissier et al.,2008; Ducrocq et al. 2008).

Figure 3: I suggest keeping only one of the four panels of Fig. 3, as the information contained in the four panels is very similar.

Only the daily SST anomaly of 14 October has been kept.

---

## Author Response (AR2)

*We thank Anonymous Referee #2 for his/her constructive comments. Our reply is in* blue *and quotes from the revised manuscript are in* purple. *Line numbers correspond to the original submission.*

**Anonymous Referee #2**

Review of the manuscript NHESS-2020-310
*The heavy precipitation event of 14–15 October 2018 in the Aude catchment: A meteorological study based on operational numerical weather prediction systems and standard and personal observations*
submitted for publication to Natural Hazards and Earth System Sciences

The Authors answered satisfactorily my previous concerns. I have only some minor comments before the paper can be accepted for publication.

**Minor remarks**

Pag. 1, lines 9-10: "as they are known to play a role in this type of hydrometeorological events".

> Corrected.

Pag. 6, lines 165-166: "The SST anomaly was more marked in the South-Western Mediterranean area with values up to 4 °C and persisted until 15 October (not shown)". Is this not shown in Fig 3?

> This is not shown for all days, but it is indeed shown for 14 October. We have replaced "(not shown)" with:

> (see fig.3 on 14 October)

Pag. 17, lines 357-358: "One could expect that a deeper trough would also be linked with stronger wind, but this was not clear on the wind correlation maps". This sentence seems in contrast with Figure 13, where it can be seen that the rainiest members are characterized by the strongest winds, as said also in the Conclusions at pag. 29, line 551: "The three rainiest members predicted the three strongest mean wind speeds blowing from the Mediterranean Sea in Leucate".

> The reviewer is right, this sentence is removed because it is misleading. The maritime wind is not directly linked with the minimum pressure inside the trough: it is linked with the pressure gradient parallel to the coast which may be strengthened by a deeper trough but depends on the location and shape of the trough. Thus, instantaneous (and not integrated over time as in fig. 13) wind correlation maps can give unclear results, especially when the location and shape of the trough is not the same in the 3 rainiest members.